



# Decadal oxygen change in the eastern tropical North Atlantic

Johannes Hahn[1], Peter Brandt[1], Sunke Schmidtko[1], Gerd Krahmann[1]

[1]GEOMAR Helmholtz Centre for Ocean Research Kiel, Düsternbrooker Weg 20, 24105 Kiel, Germany

*Correspondence to*: Johannes Hahn (jhahn@geomar.de)

**Abstract.** Repeat shipboard and multi-year moored observations obtained in the oxygen minimum zone (OMZ) of the eastern tropical North Atlantic (ETNA) were used to study the decadal change in oxygen for the period 2006-2015. At the depth of the deep oxycline (200-400 m), oxygen decreased with a rate of $-6.2 \pm 3.8$ µmol kg$^{-1}$ decade$^{-1}$, while below the OMZ core (400-1,000 m) oxygen increased by $4.1 \pm 1.7$ µmol kg$^{-1}$ decade$^{-1}$ on average. The inclusion of these decadal oxygen trends in the recently estimated oxygen budget for the ETNA

OMZ showed a weakened ventilation of the upper 400 m, whereas the ventilation strengthened homogeneously over depth below 400 m. This resulted in a shoaling of the ETNA OMZ of $-0.03 \pm 0.02$ kg m$^{-3}$ decade$^{-1}$ in density space, which was only partly compensated by a deepening of isopycnal surfaces, thus pointing to a shoaling of the OMZ in depth space as well. Shipboard, float and satellite observations of velocity and hydrography indicate different regional as well as remote changes in the circulation pattern to be responsible for

the change in the ventilation of the ETNA. The reduced ventilation in the upper 400 m may have been induced by a southward shift of the wind-driven circulation or by a change of the composition of South Atlantic Central Water. There are hints that below 400 m, latitudinally alternating zonal jets have strengthened, thus contributing to the increased ventilation. Nevertheless, temporal changes in isopycnal eddy supply or diapycnal supply (diapycnal mixing as well as diapycnal advection) cannot be excluded in having contributed to the observed

oxygen change.

## 1 Introduction

Over the past decades, tropical oceans have been subject to a conspicuous deoxygenation (Brandt et al., 2015; Stramma et al., 2008; Stramma et al., 2012) at thermocline to intermediate levels comprising the upper 700 m of the ocean. These well-documented changes manifest themselves in a profound decrease of dissolved oxygen and

a volumetric increase of open ocean tropical oxygen minimum zones (OMZs) (Stramma et al., 2010). OMZs are located in the weakly ventilated shadow zones of the ventilated thermocline (Luyten et al., 1983) in the eastern tropical Atlantic and Pacific ocean basins off the equator as well as in the northern Indian Ocean between 100 m and 900 m (Karstensen et al., 2008). Their existence predominantly results from a weak oxygen supply to these sluggish flow regimes accompanied by regionally enhanced oxygen consumption in proximity to coastal and

open ocean upwelling regions (Helly and Levin, 2004).

Among all OMZs, the open ocean OMZ of the eastern tropical North Atlantic (ETNA), with its core position around 10°N, 20°W and 400 m (Brandt et al., 2015), was observed as the region with the strongest multi-decadal oxygen decrease since the 1960s (Stramma et al., 2008). Within the recent decade, this region exhibited regionally record low oxygen concentrations well below 40 µmol kg$^{-1}$ (Stramma et al., 2009). Beyond that,

Brandt et al. (2015) found for the past decade an even stronger oxygen decrease at depth of the deep oxycline (corresponding to the upper OMZ boundary) and a weak oxygen increase below the OMZ core, superimposed on



the multi-decadal oxygen decrease. Compared to the Pacific and Indian OMZs, the open ocean ETNA OMZ –
not to be mistaken with the shallow OMZ in the Mauritanian upwelling regime (Wallace and Bange, 2004) or
episodically developing open ocean dead zones at shallow depth (Karstensen et al., 2015; Schütte et al., 2016) –
exhibits a smaller horizontal and vertical extent (300 m – 700 m) and only moderate hypoxic (oxygen
concentration < 60-120 µmol kg$^{-1}$) conditions. In particular comparison to the Pacific, the smaller basin width of
the tropical Atlantic results in quicker ventilation with younger water mass ages (Brandt et al., 2015; Karstensen
et al., 2008)). Hence, temporal changes of circulation and ventilation of the ETNA likely have a comparatively
fast impact on the regional oxygen variability.

Various processes related to anthropogenic as well as natural climate variability might have contributed to the
global multi-decadal oxygen decrease. It is generally accepted that decreasing oxygen solubility (thermal effect)
in a warming world is not the dominant driver (Helm et al., 2011) of the observed deoxygenation. Other
anthropogenically driven mechanisms for long-term changes in the oxygen distribution were identified instead
by various model studies investigating changes in biogeochemical processes with impact on the oxygen
consumption (production effect) (Keeling and Garcia, 2002; Oschlies et al., 2008) as well as changes in ocean
circulation, subduction and mixing (dynamic effect) (Bopp et al., 2002; Plattner et al., 2002; Frölicher et al.,
2009; Matear and Hirst, 2003; Schmittner et al., 2008; Cabre et al., 2015). Despite a general agreement between
different models on an anthropogenically driven and still ongoing global mean oxygen decrease, the regional
pattern of the modeled oxygen changes often does not agree with the observed trend pattern (Stramma et al.
55   2012).

Natural climate variability has been found to be an essential driver for oxygen variability on interannual to multi-
decadal time scales (Duteil et al., 2014a; Frölicher et al., 2009; Cabre et al., 2015; Stendardo and Gruber, 2012;
Czeschel et al., 2012). Alternating phases of oxygen decrease and increase might be superimposed on the
suggested anthropogenically forced multi-decadal oxygen decrease and might lead to a damping or reinforcing
of such trend on different time scales. Multi-decadal changes in circulation such as an intensification/weakening
of the tropical zonal current system, the Subtropical Cells (STC) or the Meridional Overturning Circulation
(MOC) (Chang et al., 2008; Rabe et al., 2008; Lübbecke et al., 2015) are related to changes in the ventilation and
consequently lead to varying oceanic oxygen concentrations (Brandt et al., 2010; Duteil et al., 2014a).
Nonetheless, biogeochemical and physical processes may also compensate each other leading to a damping of
the oxygen variability on interannual to decadal time scales (Duteil et al., 2014a; Cabre et al., 2015).

The pattern of the mean horizontal oxygen distribution is mainly set by the mean current field (Wyrtki, 1962). In
the tropical Atlantic, the upper ocean large-scale flow field (Schott et al., 2004; Schott et al., 2005; Brandt et al.,
2015; Eden and Dengler, 2008) comprises (i) the northern and southern STC, (ii) the western boundary current
regime as well as (iii) the system of equatorial and off-equatorial mean zonal currents (Fig. 1). The STC
describes the large-scale shallow overturning circulation, which connects of the subtropical subduction regions
of both hemispheres to the eastern equatorial upwelling regimes by equatorward thermocline and poleward
surface flow (Schott et al., 2004; McCreary and Lu, 1994). As part of the northern STC, the North Equatorial
Current (NEC) partly entrains thermocline water from the subtropics into the zonal current system south of the
Cape Verde Archipelago either via the western boundary or via interior pathways (Pena-Izquierdo et al., 2015;
Schott et al., 2004; Zhang et al., 2003), but most of this water does not reach the equator. The western boundary





current system, given by the North Brazil Current (NBC) and North Brazil Undercurrent (NBUC), acts as the major pathway for the interhemispheric northward transport of South Atlantic water. It is driven by a superposition of the Atlantic MOC (AMOC), the southern STC and the recirculation of the southward interior Sverdrup transport (Schott et al., 2004).

The zonal current system in the tropical North Atlantic drives the water mass exchange between the well-ventilated western boundary and the eastern basin (Stramma and Schott, 1999). Between 5°S and 5°N, strong mean as well as time varying zonal currents exist from the surface to intermediate depth (Schott et al., 2003; Ascani et al., 2010; Johns et al., 2014; Bunge et al., 2008; Eden and Dengler, 2008; Brandt et al., 2010; Ascani et al., 2015). They form the pronounced equatorial oxygen maximum (Brandt et al., 2008; Brandt et al., 2012) and

therewith set the southern boundary of the ETNA OMZ. Between 5°N and the Cape Verde Archipelago, wind-driven mean zonal currents are present down to a depth of about 300 m (North Equatorial Countercurrent / North Equatorial Undercurrent (NECC/NEUC) centered at 5°N and northern branch of the NECC (nNECC) centered at 9°N). In accordance with the meridional migration of the Intertropical Convergence Zone (ITCZ) these currents exhibit strong seasonal to interannual variability in their strength and position (Hormann et al., 2012; Garzoli and

Katz, 1983; Garzoli and Richardson, 1989; Richardson et al., 1992). Their connection to the western boundary seasonally alternates as well. Next to a more permanent supply of the EUC, water from the NBC retroflection is injected either into the NECC/NEUC or the nNECC. Below 300 m, latitudinally alternating zonal jets (LAZJ), occasionally referred to as latitudinally stacked zonal jets or North Equatorial Undercurrent jets, dominate the mean current field. They are a pervasive feature in all tropical oceans at intermediate depth and occur as nearly

depth-independent zonal current bands with weak mean zonal velocity of few cm s$^{-1}$ alternating eastward/westward with a meridional scale of about 2° (Maximenko et al., 2005; Ollitrault and de Verdiere, 2014; Brandt et al., 2010; Qiu et al., 2013). Within the upper 300 m, the signature of the LAZJ is often masked by a strong wind-driven circulation (Rosell-Fieschi et al., 2015). The generation of the LAZJ is not fully understood yet and different forcing mechanisms have been suggested (Kamenkovich et al., 2009; Qiu et al.,

2013; Ascani et al., 2010). Nevertheless, eddy permitting ocean circulation models partly simulate these jets leading to an improved simulated oxygen distribution in global biogeochemical circulation models (Duteil et al., 2014b) and indicating that LAZJ play an important role in ventilating the ETNA from the western boundary at surface to intermediate depth.

Two different water masses spread into the ETNA between about 100 and 1,000 m depth: North Atlantic Water
(NAW) and South Atlantic Water (SAW), which have their origin either in the North or South Atlantic (Kirchner et al., 2009). Between 150 and 500 m (corresponding to potential density layers between 25.8 and 27.1 kg m$^{-3}$), they are further subdivided in North Atlantic and South Atlantic Central Water (NACW and SACW). Both NACW and SACW exhibit almost linear $q$-$S$ relationships, where NACW is distinctly saltier than SACW. A larger fraction of SACW intrudes into the upper Central Water (UCW) layer between 150 and 300 m compared
to a smaller fraction of SACW in the lower Central Water (LCW) layer between 300 and 500 m. This layering of SACW in the upper layer above NACW in the lower layer suggests a stronger ventilation from the South Atlantic in the UCW with a vertically abrupt change in the circulation at the deep oxycline at 300 m (corresponding to a potential density 26.8 kg m$^{-3}$) (Pena-Izquierdo et al., 2015).





Below the Central Water (CW) layer, low-saline Antarctic Intermediate Water (AAIW) is found at densities
between 27.1 and 27.7 kg m$^{-3}$ (600 – 1500 m) (Stramma et al., 2005; Karstensen et al., 2008; Schmidtko and
Johnson, 2012). AAIW spreads to latitudes of about 20°N becoming less oxygenated toward north.

The recently derived observationally based oxygen budget for the ETNA (Hahn et al., 2014; Brandt et al., 2015;
Fischer et al., 2013; Karstensen et al., 2008; Brandt et al., 2010) has shed light on the different budget terms. A
balance is given between (i) oxygen supply, (ii) oxygen consumption and (iii) oxygen tendency. In the upper
350 m, mean zonal currents (NECC/NEUC, nNECC) play the dominant role for the ventilation of the ETNA and
thus, for the supply of the upper boundary of the OMZ. In the depth range of the OMZ core at about 400 m,
lateral and vertical mixing dominate the oxygen supply toward the OMZ; advection plays only a minor role.
Below the OMZ core, lateral and vertical oxygen supply weaken with depth and lateral advection becomes of
similar importance compared to the other supply terms roughly below 600 to 700 m. A recent model study
(Pena-Izquierdo et al., 2015) proposed the presence of mean vertical advection with reversed flow direction in
the UCW and LCW, which is related to two stacked subtropical cells in this depth range. Vertical advection
might play an additional role in supplying the ETNA OMZ, but so far, this process has not been considered in
the oxygen budget due to rare observational estimates of vertical velocity.

Only few processes have been investigated quantitatively (Brandt et al., 2010) that might be responsible for
driving oxygen variability on decadal to multi-decadal time scales in the ETNA. Brandt et al. (2015)
qualitatively discussed and proposed the following mechanisms: (i) decadal to multi-decadal AMOC changes;
(ii) transport variability of Indian Ocean CW entrained into the South Atlantic (variability in the Agulhas
leakage); (iii) changes in the strength of LAZJ; (iv) changes in the strength and location of the wind-driven
gyres; (v) variability of ventilation efficiency due to changes in solubility or subduction and (vi) multi-decadal
changes in the strength of Atlantic STCs. The aim of this study is to contribute to a more comprehensive
understanding of the oxygen changes during the last decade and of the dynamical processes that drive this
variability. This encompasses three major goals: (i) description of the regional pattern of the decadal oxygen
trend; (ii) determination of trends in salinity and circulation associated with the decadal oxygen trend; (iii)
discussion of implications of the decadal oxygen trend for the oxygen budget of the ETNA. The paper is
structured as follows. Section 2 describes the observational and climatological data sets used in this study.
Further, the oxygen budget and individual budget terms for the ETNA are introduced as they were mainly
derived in recent studies. In Sect. 3, data analysis and results are presented. The individual results are
comprehensively discussed in Sect. 4. Section 5 gives a short summary and conclusive remarks.

## 2 Data

This study uses a combined set of hydrographic, oxygen and velocity data, which consists of (i) repeat shipboard
and moored observations, (ii) the hydrographic climatology MIMOC, (iii) float observations provided by the
Argo program and (iv) satellite data products provided by AVISO. Further, data products of the different oxygen
budget terms are used, which were derived in recent studies. Acquisition and processing of hydrographic,
oxygen and velocity data from repeat shipboard and moored observations are described in Sect. 2.1 and 2.2. The
climatological data set MIMOC (Schmidtko et al., 2013) is introduced in Sect. 2.3. Section 2.4 presents the Argo
Float data set (Roemmich et al., 2009) and Sect. 2.5 introduces the sea surface geostrophic velocity data derived





from satellite altimetry. A detailed description of the oxygen budget and the oxygen budget terms estimated in recent studies is given in Sect. 2.6.

### 2.1 Repeat ship sections along 23°W

Hydrographic, oxygen and velocity data were obtained during repeated research cruises carried out mainly along 23°W in the ETNA in the period from 2006 to 2015 (1999 to 2015 for velocity; see Table 1 for details). The data set is an update of the 23°W ship section data used in Brandt et al. (2015). In the present study, data records from 4°N to 14°N (ETNA OMZ regime) were taken into account.

For an individual ship section along 23°W, hydrographic and oxygen data were acquired during regular CTD 160 casts, which were performed typically on a uniform latitude grid with half degree resolution. Velocity data was acquired with different realizations of Acoustic Doppler Current Profilers (ADCPs), where vessel-mounted ADCPs (vm-ADCP) recorded continuously throughout the section and a pair of lowered ADCPs (l-ADCP), attached to the CTD rosette, recorded during regular CTD casts.

Based on the acquired data sets, meridional sections of hydrography, oxygen and velocity were calculated for 165 each ship section on a homogeneous latitude-depth grid with a resolution of 0.05 ° and 10 m, respectively. Details on the methodology as well as on measurement errors are given in Brandt et al. (2010). For a single research cruise the accuracy of hydrographic and oxygen data was assumed to be generally better than 0.002 °C, 0.002 and 2 $\mu$mol kg$^{-1}$ for temperature, salinity and dissolved oxygen, respectively. The accuracy of 1-h averaged velocity data from vm-ADCPs and single velocity profiles from l-ADCP was better than 2–4 cm s$^{-1}$ and 170 5 cm s$^{-1}$, respectively.

### 2.2 Moored observations

Multi-year moored observations (2009 – 2015) were performed in the ETNA to record hourly to interannual oxygen variability at three different positions: 5°N/23°W, 8°N/23°W and at about 11°N/21°W (see Table 2 for details about available data). All moorings were equipped with oxygen (AADI Aanderaa optodes of model types 175 3830 and 4330) and CTD sensors (Sea-Bird SBE37 microcats), which were attached next to each other on the mooring cable to allow an appropriate estimate of the dissolved oxygen on density surfaces. At every mooring site, eight evenly distributed optode/microcat combinations were installed in the depth range between 100 to 800 m delivering multi-year long oxygen time series with a temporal resolution of up to 5 min.

Moorings were serviced and redeployed generally every 18 months. In order to achieve the highest possible 180 long-term sensor accuracy, optodes and microcats were carefully calibrated against oxygen and CTD measurements (from a CTD/O$_2$ unit) by attaching them to the CTD rosette during regular CTD casts immediately prior to and after the mooring deployment period. Optodes were additionally calibrated in the laboratory on board to expand the range of reference calibration points. Details on the optode calibration methodology are given in Hahn et al. (2014). The root mean square error of all temperature and salinity 185 measurements (microcats) as well as dissolved oxygen measurements (optodes) was about 0.003 °C, 0.006 and 3 $\mu$mol kg$^{-1}$, respectively.



### 2.3 MIMOC climatology

The monthly, isopycnal and mixed-layer ocean climatology (MIMOC) (Schmidtko et al., 2013) was used as a reference mean state in order to (i) perform a water mass analysis and (ii) quantify the temporal evolution of salinity anomalies in the tropical Atlantic. The climatological fields of potential temperature and salinity were interpolated on respective density layers and mean $\theta$-$S$ characteristics were defined for the two predominant water masses NAW and SAW found in the tropical Atlantic.

### 2.4 Argo float data set

All available Argo data was used (http://dx.doi.org/10.13155/29825). Argo float data is collected and made freely available by the international Argo project and the national programs that contribute to it. Here, Argo float data flagged by the data centers was removed and delayed mode data was used for the profiles as available. Then, the remaining good data was used in this study to quantify the temporal evolution of salinity anomalies in the tropical Atlantic.

### 2.5 Surface geostrophic velocity from altimeter products

Surface geostrophic velocity data derived from satellite altimetry was used in this study to estimate the decadal change of the near-surface circulation and ventilation of the ETNA. The altimeter product was produced by Ssalto/Duacs and distributed by Aviso, with support from Cnes (http://www.aviso.altimetry.fr/duacs/). Absolute geostrophic velocity data was taken from the delayed time global product *MADT* (Maps of Absolute Dynamic Topography) given in the version *all sat merged*. The data was extracted for the box 4°N – 14°N / 35°W – 20°W for the data period January 1993 – September 2015 (processing date: 27-Nov-2015).

### 2.6 Oxygen budget terms

The oceanic oxygen distribution is governed on the one hand by oxygen supplying processes such as physical transport or photosynthetic oxygen production and on the other hand by oxygen consumption driven by biological respiration or remineralisation of sinking organic matter (Karstensen et al., 2008; Brandt et al., 2015). Supply and consumption are usually not in balance, as for the ETNA the oxygen consumption is about 10 - 20 % larger than the supply processes resulting in the aforementioned multi-decadal oxygen trend (Brandt et al., 2010; Hahn et al., 2014; Karstensen et al., 2008)). Hence, the oxygen budget is balanced, when including the oxygen tendency additionally to oxygen supply and consumption. However, to analyze temporal changes in the budget terms, which cause a change in the oxygen trend from decadal to multi-decadal time scales (Brandt et al., 2015), a time-varying oxygen budget has to be taken into account. Following Hahn et al. (2014) and Brandt et al. (2015), we formulate the non-steady state depth-dependent oxygen budget for the ETNA over the latitude range 6°N - 14°N as:

$$\left[\underset{[1]}{\underline{\partial_t O_2}}\right]^{(i)} = \underset{[2]}{\underline{aOUR}} + \underset{[3]}{\underline{K_e \partial_{yy} O_2}} + \underset{[4]}{\underline{K_\rho \partial_{zz} O_2}} + \left[\underset{[5]}{\underline{R_{O_2}}}\right]^{(i)}, \tag{1}$$

where the superscript on both sides of the equation denotes the time variation with respect either to the decadal *(i=1)* or multi-decadal *(i=2)* oxygen trend. This oxygen budget equation (1) describes the components contributing to the changes of the oxygen concentration all of which are given in µmol kg$^{-1}$ yr$^{-1}$. Term [1] ($\partial_t O_2$)



on the left hand side of the equation marks the observed temporal change of oxygen (oxygen tendency). While the multi-decadal oxygen trend was already considered in Hahn et al. (2014), the estimate of the decadal oxygen trend and its inclusion in the oxygen budget is a central part of this study (details on the calculation are given in Sect. 3.2 and 3.4). Term [2] ($aOUR$) defines the oxygen consumption, which has been determined in Karstensen

et al. (2008) following an approach that relates the apparent oxygen utilization (AOU) to water mass ages. Term [3] ($K_e \partial_{yy} O_2$) is the eddy-driven meridional oxygen supply representing the divergence of the meridional oxygen flux due to isopycnal eddy diffusion (Hahn et al., 2014). Term [4] ($K_\rho \partial_{zz} O_2$) defines the diapycnal oxygen supply representing the divergence of the diapycnal oxygen flux due to turbulent mixing (Fischer et al., 2013). Term [5] includes all other oxygen supply mechanisms and is calculated as the residual oxygen supply

based on terms [1] to [4]. For the sake of completeness, [5] is nonspecifically considered as a composite of mean advection, zonal eddy diffusion as well as submesoscale processes. We follow the argumentation given in Hahn et al. (2014): by considering the meridional structure of the eddy-driven meridional oxygen supply (term [3] in our Eq. (1)) as well as the horizontal oxygen distribution (see also Fig. 13 therein), they argued for a major contribution of the mean zonal advection in the upper 350 m, while zonal eddy diffusion and mean meridional

advection have only minor effect and submesoscale processes are not assumed to affect the oxygen distribution well below the base of the mixed layer (Thomsen et al., 2016). Note that vertical advection, which was recently proposed in a model study (Pena-Izquierdo et al., 2015) to be present as part of two stacked subtropical cells, was so far not considered in the observationally based oxygen budget.

Given the data set in this study, a complete analysis of the temporal change of the consumption and supply

terms, which ultimately is responsible for the change in the oxygen trend, cannot be performed due to data coverage and respective uncertainty reasons. Nevertheless, the decadal and multi-decadal oxygen trends ($[\partial_t O_2]^{(1)}$ and $[\partial_t O_2]^{(2)}$) are applied separately in the oxygen budget. All other directly calculated terms (consumption, meridional eddy supply and diapycnal supply) are kept time-invariant, while the residual oxygen supply is calculated for the respective time periods ($R_{O_2}^{(1)}$ and $R_{O_2}^{(2)}$) based on the two oxygen trends. This ad hoc

approach is used to particularly discuss zonal advection as a potential driver for the change in the oxygen trend.

### 3 Data analysis and results

In this section, results will be shown from the combined analysis of moored, shipboard and float observations in the tropical Atlantic with a particular focus on the ETNA OMZ in order to investigate and quantify decadal changes of oxygen as well as their correlation with changes in salinity. Changes in the velocity field were

estimated based on repeat shipboard observations as well as satellite observations obtained mainly in the ETNA OMZ, although a direct relation among velocity and oxygen or salinity may not be derived for the respective time scales. Eventually, the oxygen budget of the ETNA was reanalyzed and revised from Hahn et al. (2014) with respect to both decadal and long-term oxygen changes.

### 3.1 Mean state in the ETNA

The mean state 23°W sections of oxygen, hydrography and zonal velocity were calculated between 4°N and 14°N (Fig. 2a to 2c) based on all uniformly gridded 23°W sections as obtained from the repeat shipboard observations (cf. Sect. 2 and Table 1). For the depth range 100 - 1,000 m, the averaging of all sections yielded an



average standard error of the mean fields (which is considered to result dominantly from oceanic variability) of about 1.8 µmol kg⁻¹, 0.06 °C, 0.009 and 1.1 cm s⁻¹ for oxygen, temperature, salinity and zonal velocity,
respectively.

Along the 23°W section, the core of the deep OMZ is located at 10°N and at 430 m depth with a minimum oxygen concentration of 41.6 µmol kg⁻¹ (Fig. 2a). Between 100 and 250 m, pronounced oxygen maxima at 5°N and 8°N-9°N coincided well with the core positions of the near-surface NECC and nNECC (Fig. 2c). Similar patterns could not be observed for salinity distribution (Fig. 2b). However, the largest meridional salinity
gradient was found in the CW layer (25.8 – 27.1 kg m⁻³) around 10°N and coincides with the band of the near-surface eastward nNECC and mirrors the transition zone from SAW to NAW, i.e. from low salinity close to the equator to high salinity in the northern part of the section.

For the ETNA, the transition from SAW to NAW is well reflected in the θ-S diagram (Fig. 2d). θ-S characteristics were derived for particular latitudes based on the mean hydrographic ship section along 23°W
(Fig. 2b, temperature not shown). Characteristics for pure NAW and SAW were defined from the MIMOC climatology for the areas [25°N-30°N / 60°W-10°W] and [5°S-0°N / 40°W- 0°E], respectively. Following Rhein et al. (2005) a simple water mass analysis was evaluated taking into account NAW and SAW (not shown). This exhibited a strong spreading of SAW towards the north in the upper 300 m (UCW layer) with a SAW fraction of 0.9 close to the Cape Verde Archipelago at 13°N. Below 300 m (LCW and IW layer), SAW and NAW occurred
with equal contribution, while NAW had its southernmost extension at about 550 m with a fraction of 0.2 at 6°N.

### 3.2 Variability of oxygen and salinity on interannual to decadal time scales

The mean states of oxygen and hydrography based on shipboard observations along 23°W were examined in the previous section. Here we present the temporal variability of oxygen and salinity obtained from shipboard and moored observations of the most recent decade and investigate the correlation between these two variables.
Decadal trends were derived at the aforementioned mooring positions as well as for the 23°W section between 4°N and 14°N.

Figures 3a/3d, 4a/4d and 5a/5d show the combination of repeated 100 – 1,000 m profiles of oxygen and salinity obtained from shipboard observations along 23°W at 5°N, 8°N and 11°N together with depth-interpolated 10-day low pass filtered time series of oxygen and salinity obtained from nearby moored observations between
100 and 800 m (cf. Table 2). Corresponding oxygen and salinity anomalies (Fig. 3b/3e to Fig. 5b/5e) were calculated by (i) interpolating the moored and shipboard observations on a regular density grid (grid spacing 0.01 kg m⁻³), (ii) subtracting the respective mean profiles in density space obtained solely from shipboard observations (Fig. 3c/3f to Fig. 5c/5f), and (iii) projecting the calculated anomalies back onto depth grid (by applying an average depth-density relation) in order to obtain a more intuitive presentation in depth space.

While the long-term mean 23°W oxygen section showed a minimum oxygen concentration of 41.6 µmol kg⁻¹ (see Sect. 3.1), individual CTD casts from shipboard observations throughout the past decade regularly exhibited minimum oxygen concentrations well below 40 µmol kg⁻¹ in the core of the deep OMZ between 300 and 700 m (cf. Fig. 5a; see also Stramma et al. (2009)). An absolute minimum of 36.5 µmol kg⁻¹ was observed at 12.5°N, 23°W at 410 m during RV *Maria S. Merian* cruise MSM22 in November 2012. The combined analysis of
shipboard and moored observations reveals a remarkable oxygen change at the latitude of the ETNA OMZ core



throughout the past decade (Fig. 5b/5e). Oxygen strongly decreased in the upper 500 m and increased below, with salinity generally showing opposite trends. Decadal changes were less pronounced south of the OMZ core (Fig. 3b/3e and 4b/4e) accompanied by an increased intraseasonal to interannual variability.

Changes in the spatial oxygen distribution go along with changes in the OMZ core position. Here, the vertical and meridional position of the OMZ core was estimated for every ship section by taking the center of the 1%-area of lowest oxygen found between 8°N and 14°N in depth (400 – 460 m) and density space (27.00 – 27.06 kg m$^{-3}$), respectively. Decadal trends with 95% confidence were then calculated for depth, density and latitude of the OMZ core in order to derive the vertical and meridional migration of the OMZ core (Fig. 6). We found a migration of the OMZ core position in density space toward lighter water (-0.03 ± 0.02 kg m$^{-3}$ decade$^{-1}$), whereas an upward migration in depth space could not significantly be shown (-8 ± 34 m decade$^{-1}$). No significant meridional migration of the OMZ core was observed both in depth and density space (about 0.7 ± 2.6 ° decade$^{-1}$).

In order to relate changes in oxygen to changes in physical ventilation processes, we computed the correlation of oxygen and salinity on isopycnal surfaces from moored and shipboard observations (Fig. 7). As we consider only long-term variability, the correlation was computed from the 90-day median of the mooring time series. Both individual observational data sets show a strong negative correlation below the deep oxycline and south of the OMZ core - a regime with a pronounced positive gradient in salinity and negative gradient in oxygen on isopycnal surfaces in northeast direction (cf. Fig. 2 in this study, Kirchner et al. (2009); Pena-Izquierdo et al. (2015)). North of the OMZ core, the correlation is positive, which agrees well with the positive oxygen and salinity gradient in northward direction in this regime. Note that above the deep oxycline (upper 300 m) the correlations obtained from moored and shipboard observations partly disagree with each other, which might be due to generally larger variability and the different time periods covered by both observational methods.

Decadal oxygen and salinity trends on isopycnal surfaces (Fig. 8) were derived from the combined observational data set (as shown in Fig. 3 to 5) using a weighted linear regression scheme. A single ship section was weighted similar to 30 days of moored observations. Over the last decade at all three mooring positions, oxygen decreased in the upper 300 to 400 m and stayed constant or increased below (Fig. 8a). Salinity increased between 200 and 400 m at 8°N and 11°N (Fig. 8b), while a salinity decrease or no salinity change at all was found between 500 and 800 m for all latitudes.

In order to investigate the meridional section of the decadal oxygen and salinity trends, solely shipboard observations were used (Fig. 9). Therefore linear regressions were calculated (using a 95% confidence level to determine their significance) at each grid point in latitude-density space. The resulting sections were subsequently projected onto depth grid. The section of the decadal oxygen trend (Fig. 9a) reveals coherent large-scale patterns of oxygen decrease and oxygen increase. Even though only a part of the local trends is statistically significant, the spatial coherence of the patterns suggests that these trends are robust. Note also that significant patterns were larger than twice the smoothing and interpolation scale of the individual ship sections. Strongly decreasing oxygen concentration with up to -2 µmol kg$^{-1}$ yr$^{-1}$ was found in the latitude range from 6°N to 14°N and at a depth range 200 to 400 m (on average -0.62 ± 0.38 µmol kg$^{-1}$ yr$^{-1}$). At about 100 m such a decrease was found throughout the whole section (4°N to 14°N). Between 400 and 1,000 m, oxygen was found to increase with an average magnitude of about 0.41 ± 0.17 µmol kg$^{-1}$ yr$^{-1}$ between 6°N and 14°N. Strictly, two maxima of


oxygen increase were observed below the OMZ core depth (500 – 700 m) in the latitude ranges 7°N-10°N and 12°N-13°N, respectively, with a weaker oxygen increase in between.

The section of the decadal salinity trend for the past decade (Fig. 9b) shows a salinity increase in the latitude range between 6°N – 12°N at a depth range of 200 to 350 m as well as at shallower depths (100 – 300 m) between 12°N – 13°N, where a local maximum in salinity increase was observed at 12.5°N. Between 350 and 340 800 m, salinity decreased in two latitude bands (7°N - 10°N and 12°N - 14°N) as well as increased (though not significant) in adjacent latitude bands (5°N – 7°N and 10°N – 12°N).

Patterns of the decadal change in salinity along 23°W were found to be negatively correlated with those in oxygen in the density layers of LCW and IW (Fig. 10b and 10c). In particular, the strongest negative correlation occurred in the latitude band 6°N – 8°N weakening northwards and no correlation at all was present north of the 345 OMZ core in the latitude band 12°N – 14°N. This relation qualitatively agrees well with the observed correlation patterns of oxygen and salinity (Fig. 7).

Changes in salinity over the past decade were further investigated for the whole tropical Atlantic at two characteristic density surfaces (26.8 kg m$^{-3}$ and 27.2 kg m$^{-3}$) based on Argo float observations (Fig. 11). Anomalies of salinity were calculated with respect to the mean state given by the MIMOC climatology (see Sect. 350 2.3). At the shallower density layer (26.8 kg m$^{-3}$), salinity generally increases between 10°S and 10°N. Here, a propagation of positive salinity anomalies was observed from the tropical South Atlantic to the tropical North Atlantic throughout the past decade (cf. Kolodziejczyk et al. (2014) for shallower depths). Further north, a negative salinity trend was found between about 10°N and 20°N, and a positive trend was found north of 20°N. At the deeper density layer (27.2 kg m$^{-3}$), the northward propagation of positive salinity anomalies from the 355 tropical South Atlantic was less pronounced.

### 3.3 Circulation variability in the ETNA

A change in the large-scale circulation is possible source for the decadal changes in oxygen and salinity shown in the previous section. Such circulation changes would likely precede changes in the discussed tracers (cf. Brandt et al. (2012)) and several-decade-long observations would be necessary in order to attribute such a 360 relationship. Here, we are far away of quantitatively linking circulation variability to decadal changes in oxygen and salinity. Nevertheless, in the following we present observed changes in the circulation in order to discuss the potential impact on the hydrographic and oxygen distribution in the ETNA.

The decadal change in the depth of the density surfaces (isopycnal heave tendency, Fig. 12) was derived from all hydrographic shipboard observations carried out along 23°W between 2006 to 2015. Density surfaces deepened 365 at latitudes 7°N – 12°N below 100 m (3.3 ± 3.6 m decade$^{-1}$ at the OMZ core depth) and shoaled north and south of it, which points to a change in the geostrophic circulation. This change goes along with an increased stratification within the OMZ and a decreased stratification north and south of it. Repeat shipboard ADCP and hydrographic observations were subsequently used in order to investigate the decadal change in the zonal velocity. Zonal velocity sections derived from a geostrophic approach as well as from shipboard ADCP 370 observations were very similar both reasonably representing the wind-driven near-surface currents (NECC and nNECC) as well as the LAZJ at depth. In the following, only shipboard ADCP observations are shown.





Zonal velocity from shipboard ADCP observations was estimated along 23°W as an average for the periods 1999-2008 and 2009-2015 (Fig. 13a and 13b) to determine the decadal change. A one-tailed t-test (directional hypothesis) was applied to evaluate the significance (95% confidence) of the difference of the means for the two periods (Fig. 13c to 13e). Zonal velocity didn't change for most parts of the section (80% - 90%), but two latitudinal regimes were identified with significant difference between both periods. First, the circulation south of the Cape Verde Islands reversed, where, in contrast to the first period, in the latter period and over the whole depth range eastward and westward currents were observed at latitudes 12°N-13°N and 13°N-14°N, respectively. Second, the eastward jet, centered at about 9°N, widened at depths below about 300 m from the first to the latter period, going along with a narrowing of the westward jet north of it.

In order to investigate the variability of the wind-driven near-surface currents, we used altimetry based surface geostrophic velocity observations provided by AVISO (Sect. 2.5). The two main variability patterns of the surface circulation in the tropical North Atlantic can be described by the first complex EOF (empirical orthogonal function) mode of the zonal geostrophic velocity anomaly, where the real part of the EOF pattern mimics the meridional migration of the NECC and the imaginary pattern reflects the variability in its strength (Hormann et al., 2012). We applied an EOF analysis to the grid point wise filtered (mean and seasonal cycle removed and subsequently 2-year-low-pass-filtered) time series of AVISO zonal geostrophic velocity for the region 35°W-20°W and 4°N-14°N in order to capture the interannual to decadal variability of the NECC/nNECC throughout the past decades (Fig. 14). The first two EOF modes (Fig. 14a and 14b) explained 23% and 14% of the total variance of the filtered time series and can be considered similarly to the real and imaginary pattern of the complex EOF as given in Hormann et al. (2012). In relation to the mean zonal geostrophic velocity (additionally shown in Fig. 14a and 14b), the first EOF describes maximum (out-of-phase) variability of the zonal geostrophic velocity north and south of the NECC core latitude at about 6°N (capturing a meridional migration of the NECC), whereas the second EOF describes maximum variability at about the core latitude of the NECC (capturing an intensification or weakening of the NECC). The principal component (PC) corresponding to the first EOF mode (first and second PC shown in Fig. 14c) declined throughout the past decade, which points to a southward migration of the NECC. The PC corresponding to the second EOF mode revealed only shorter-term (interannual) variability in the strength of the NECC. Reconstructing the zonal geostrophic velocity anomaly out of the first two EOFs and the corresponding PCs with subsequent averaging in the boxes [35°W – 20°W, 4°N – 6°N] and [35°W – 20°W, 6°N – 10°N] (Fig. 14d) shows that eastward velocity increased in the latitude band 4°N - 6°N and decreased in the latitude band 6°N - 10°N throughout the past decade.

### 3.4 Decadal and long-term changes in the oxygen budget of the ETNA OMZ

The multi-decadal oxygen trend (1972 – 2008) was discussed in the oxygen budget of the ETNA OMZ in Hahn et al. (2014) (see also Sect. 2.6). Here, the oxygen trend observed for the past decade was additionally included (Fig. 15) in order to discuss temporal changes in the oxygen budget. The depth profile of the decadal oxygen trend was calculated as an average between 6°N and 14°N of the 23°W section (Fig. 9a). During the period 1972 – 2008, an oxygen decrease was observed over the whole depth range (here: shown for 130 – 800 m), whereas for the past decade (2006-2015) an oxygen decrease / increase was found in the depth range 150 – 400 m / 400 – 800 m. Other terms in the oxygen budget, namely oxygen consumption, meridional eddy supply and diapycnal





supply were calculated as described in Brandt et al. (2015) as well as in corresponding studies of Karstensen et al. (2008), Fischer et al. (2013) and Hahn et al. (2014).

The vertical profiles of the decadal (2006 - 2015) and multi-decadal (1972 - 2008) oxygen trend, respectively, were independently applied in the oxygen budget (Fig. 15). Other terms such as consumption, meridional eddy
supply and diapycnal supply were kept time-invariant. The change in the oxygen trend between the two periods was projected onto the residual oxygen supply (cf. Eq. (1)), which is mainly attributed to the advective supply due to zonal jets ventilating the eastern basin from the oxygenated western boundary (Brandt et al., 2015). At first glance, this projection seems like a rather strict assumption, though the temporal variability in the oxygen budget shall be discussed exemplarily using the residual supply without loss of generality. According to the two
oxygen trend profiles, two different residual supply profiles resulted for the periods 1972 - 2008 and 2006 - 2015, respectively. Both residuals show a strong advective supply in the upper 350 m including a steep drop-off and subsequently a rather homogeneous supply between 400 and 800 m. In comparison to the earlier period 1972 - 2008, the residual supply in the depth range below 400 m homogeneously doubled in magnitude for the recent decade (2006 - 2015). The strong advective supply in the upper 350 m only changed marginally
and shifted to a slightly shallower depth in the past decade (2006 - 2015) being barely significant with regard to the large confidence intervals.

## 4 Discussion

A comprehensive observational data set consisting of moored, shipboard, float and altimetry observations from the past two decades in the ETNA has been used to study the variability of hydrography, oxygen and circulation
with a focus on the decadal change as well as its respective implication for the oxygen budget. Shipboard observations give a detailed view on the regional pattern of the decadal change, whereas moored and float observations additionally capture variability on interannual and shorter time scales.

### 4.1 Variability of oxygen and salinity in the ETNA

Shipboard and moored observations between 4°N and 14°N along 23°W revealed a decadal change of oxygen
with a strong decrease (Fig. 3 to 5, 8a and 9a) at depth of the deep oxycline between 200 and 400 m (-$6.2 \pm 3.8$ µmol kg$^{-1}$ decade$^{-1}$) and a large-scale increase below ($4.1 \pm 1.7$ µmol kg$^{-1}$ decade$^{-1}$). This vertically oriented dipole pattern is centered approximately at depth of the OMZ core and is consistent with the observed upward migration of the OMZ core in density and depth space throughout the past decade (Fig. 6). Strictly, shipboard observations revealed a significant shoaling of the OMZ core only in density space. However, the
average vertical migration in density space (-$0.03 \pm 0.02$ kg m$^{-3}$ decade$^{-1}$) was too large to be compensated by the isopycnal heave (Fig. 12; $3.3 \pm 3.6$ m decade$^{-1}$) when considering the observed mean background vertical density gradient of about $1.2 \times 10^{-3}$ kg m$^{-3}$ per m (not shown). This inconsistency is compensated by assuming an additional upward migration of the OMZ core in depth space, which we estimated to be -$22 \pm 17$ m decade$^{-1}$.

While the decadal oxygen change is characterized by a vertical dipole pattern, the multi-decadal oxygen trend
(period 1972 – 2013), analyzed for the tropical and subtropical Atlantic along 23°W by Brandt et al. (2015), shows a large-scale moderate oxygen decrease for the whole ETNA OMZ regime. Maximum rates of about -5 µmol kg$^{-1}$ decade$^{-1}$ therein were substantially smaller than maximum rates of oxygen decrease (-20 µmol kg$^{-1}$





decade$^{-1}$) found in this study for the recent decade (Fig. 9a). At the OMZ core depth and below, observed opposite oxygen trends on decadal and multi-decadal time scales suggest that different mechanisms may act in parallel and drive these oxygen changes. Note that the local maximum of the multi-decadal oxygen decrease (Brandt et al., 2015) is at around 20°N and 450 m (-7 µmol kg$^{-1}$ decade$^{-1}$) indicating a northward shift of the ETNA OMZ likely associated with a similar shift in the circulation pattern. In contrast, the observed upward migration of the ETNA OMZ in the recent decade might be understood as the consequence of a decreased ventilation in the upper 400 m and an increased ventilation below.

Oxygen variability was strongly related to salinity variability in the ETNA at depth of the deep oxycline and below (Fig. 7). Both variables were found to be negatively correlated south of the OMZ core position, while a weak positive correlation was found north of it. This reflects the supply of the OMZ with fresher SAW from the southern hemisphere as well as saltier NAW from the northern hemisphere via different ventilation pathways (Brandt et al., 2015; Pena-Izquierdo et al., 2015; Kirchner et al., 2009). Nevertheless, we cannot exclusively link decadal changes in hydrography to decadal changes in the water mass composition in order to explain changes in oxygen, as source waters may change its $\theta$-$S$ characteristics as well. Instead, a straightforward approach is the analysis of salinity variability to detect both a change in the ventilation as well as a change in the source water masses.

Decadal change in oxygen along 23°W was negatively correlated with the decadal change in salinity in the LCW and IW layer (below 300 m) south of the OMZ core position (Fig. 9 and 10). In the deep oxycline, the strong oxygen decrease was accompanied by a salinity increase, whereas below 400 m the large-scale oxygen increase was accompanied partly by a salinity decrease. Argo float observations from the tropical and subtropical Atlantic (Fig. 11) revealed a large-scale increase in salinity between 10°S to 10°N. This increase was pronounced at depth of the deep oxycline (around density surface 26.8 kg m$^{-3}$), while in the IW layer (here shown for density surface 27.2 kg m$^{-3}$) salinity increase was weaker and less homogeneous.

Different mechanisms can be responsible for the decadal change in salinity. The Argo float observations reflected a northward migration of positive salinity anomalies advected from the southern to the northern hemisphere along the western boundary. These positive salinity anomalies in the South Atlantic might have resulted from (i) a change in the Agulhas leakage (Kolodziejczyk et al., 2014; Hummels et al., 2015; Biastoch et al., 2009) or (ii) a change in the subduction rate in the subtropical South Atlantic (Liu and Huang, 2012). Both scenarios might have gone along with a change in the water mass age and thus with a change in the ventilation time of the ETNA. However, (iii) global warming could be another cause for the observed change in the salinity distribution either induced by a vertical displacement of isopycnal surfaces in the tropical Atlantic or by a horizontal displacement of isopycnal surfaces in the subduction regimes in the subtropics. Note that despite a negative salinity trend observed between 10°N and 20°N, an increased ventilation from the subtropical North Atlantic might have also contributed to the salinization south of 10°N.

### 4.2 Changes in the circulation drive changes in oxygen and salinity

The generally negatively correlated decadal changes in oxygen and salinity (Fig. 9 to 11) in the ETNA along 23°W (Fig. 12) in the LCW and IW layer suggest a change in the circulation pattern of the tropical Atlantic (200 – 400 m: weakened ventilation; below 400 m: intensified ventilation). This was further corroborated by the





shoaling of the ETNA OMZ (see also Sect. 4.1) as well as by the deepening of isopycnal surfaces in the OMZ regime between 7°N and 11°N, while a shoaling of isopycnal surfaces was observed north and south of it. A more detailed discussion about circulation changes is given in the following.

In the deep oxycline (200 – 400 m), advection of SAW from the western boundary is an important factor for the 490 ventilation of the ETNA (see Fig. 2 and Fig. 11 in this study, and Hahn et al. (2014); Brandt et al. (2015); Pena-Izquierdo et al. (2015)). Thus, it is likely that interannual to decadal variability in the position and strength of the largely wind-driven currents in the top few hundred meters might have caused the changes in oxygen and salinity on isopycnal surfaces. Repeated 23°W sections didn't show a large-scale change in the strength of the zonal velocity (Fig. 13). However, the analysis of altimetry based surface zonal geostrophic velocity revealed a 495 strengthening of the eastward velocity between 4°N and 6°N (NECC) and a weakening between 6°N and 10°N (nNECC) from 2006 to 2015 (Fig. 14d), which is associated with a southward shift of this zonal velocity band. These circulation changes that are typically forced by the large-scale wind-stress curl result in a weakened ventilation of the ETNA from the western boundary in the latitude range of the OMZ, which is in agreement with the observed decadal change pattern of oxygen and salinity in this depth range.

A distinct change in the current field was found directly south of the Cape Verde Islands (Fig. 13). The band of eastward velocity shifted toward south (from 14°N to 12°N-13°N) and a band of weak westward velocity was observed instead around 13°N-14°N during the more recent period. This can be associated with a southward shift of the Cape Verde Current (CVC, Pena-Izquierdo et al. (2015)), which is thought to predominantly advect NAW. However, it remains uncertain, whether a southward shift of the CVC has contributed to a change in the ratio of 505 NAW and SAW, and to a (de)oxygenation in the upper 400 m of the water column.

In addition to the decadal change, moored observations have also revealed pronounced oxygen and salinity variability on interannual time scales. Associated interannual to decadal variability of the wind-driven circulation in the tropical Atlantic is predominantly driven by atmosphere-ocean interactions related to Atlantic climate modes, namely the Atlantic Zonal Mode (AZM) and Atlantic Meridional Mode (AMM) (Chang et al., 2006; 510 Joyce et al., 2004; Marshall et al., 2001; Zhu et al., 2012). Servain (1991) introduced the AZM as the equatorial sea surface temperature (SST) anomaly and defined the AMM as the anomalous inter-hemispheric SST gradient with negative AMM corresponding to a negative/positive SST anomaly in the Northern/Southern Hemisphere. A negative AZM or a positive AMM was found to be associated with a northward shift and a strengthening of the NECC (Hormann et al., 2012). Although both climate modes are not correlated with each other, the authors 515 report one of the strongest negative/positive AZM/AMM in the year 2005 and a phase of intense positive/negative AZM/AMM in the years 2008-2010 (particularly see Foltz et al. (2012) for the strong negative AMM in 2009). These phases are related to an anomalous strengthening as well as northward shift (in the earlier period) and a weakening as well as southward shift (in the latter period) of the NECC, respectively. Similar analyses in our study showed a still persistent anomalous southern position of the NECC in recent years (Fig. 14) 520 – superimposed by interannual variability in NECC strength and position - resulting in almost a full decade of weakened ventilation of the upper 300 m in the ETNA.

The AMM has recently been shown to be interrelated to the Guinea Dome variability (Doi et al., 2010), where a negative AMM in boreal spring is connected to a southward shift of the ITCZ as well as a weaker Guinea Dome. The aforementioned multi-annual negative AMM phase might be associated with a comprehensive weakening of





the near-surface circulation on interannual to decadal time scales, which contributed to a weakened ventilation
and subsequently to the average negative oxygen trend in the deep oxycline in the recent decade.

Below 400 m, the large-scale oxygen increase along 23°W pointed to an intensification of the ventilation of the
ETNA throughout the past decade. A likely reason is a strengthening of the LAZJ during the past decade
compared to the decade before, subsequently causing the oxygen increase. Time coverage of velocity data was
generally too sparse and too short to conduct a profound analysis of the interannual to decadal velocity
variability (neither for Argo (Rosell-Fieschi et al., 2015) nor ship section data [*this study*]). However, we found a
slight broadening and intensification of the eastward jet at about 9°N and a southward shift of the CVC from
13.5°N to 12.5°N (Fig. 13) for the recent one and a half decades. This change in the velocity field is reasonably
reflected in the decadal change pattern of oxygen and salinity. The eastward jet at 9°N advected more SAW
(fresh, oxygenated) from the western boundary. In addition, the southward shifted CVC may have led to an
advection of more oxygenated water at 13°N. If we assume a time invariant oxygen consumption, higher
oxygenated water supplied by eastward jets agrees well with the only little loss up to little gain of oxygen found
in the recirculated water of adjacent westward current bands at 7°N and 11°N. This result is not contradictory to
the deoxygenation at 200 to 400 m depth: LAZJ are low baroclinic mode currents (e.g. Qiu et al., 2013), which
are superimposed on the strong wind-driven current field of the upper 350 m (Rosell-Fieschi et al., 2015). The
comparatively small temporal variability of the LAZJ is assumed to have almost no impact on the oxygen and
water mass variability close to the surface.

Hahn et al. (2014) show that lateral mixing is a dominant driver in supplying the OMZ below 400 m through its
meridional boundaries. Tropical instability waves, acting at the southern boundary at about 5°N, have varied in
strength throughout the past decade potentially leading to a temporal change in the ventilation. They were
anomalously weak in the years 2006-2008 and strengthened afterward in the years 2009-2010 (Perez et al.,
2012). This may explain the pronounced interannual variability at 5°N in comparison to 11°N (Fig. 3 to 5); a
possible effect of decadal changes in the strength of the lateral mixing on the oxygen variability cannot be
addressed with the available data. However, an increase of the meridional eddy diffusivity would have also
affected the upper 400 m. We suspect that an increasing meridional eddy diffusion on interannual to decadal
time scales would have projected the band structure of the eddy-driven meridional oxygen supply (latitudinally
alternating loss and gain of oxygen; cf. Fig. 13 in Hahn et al. (2014)) into the decadal change pattern of oxygen.
Thus, an exclusive intensification of the lateral eddy diffusion does not reasonably fit with the observed
variability patterns.

**4.3 Oxygen budget for decadal and multi-decadal oxygen changes**

A time varying oxygen budget of the ETNA (6°N-14°N) was evaluated by including both the decadal and multi-
decadal oxygen trend (Fig. 15) in order to discuss potential changes in the ventilation terms (lateral mixing,
diapycnal mixing, advection). The following discussion shall strengthen the argumentation on decadal changes
in the ventilation given in the last section.

Brandt et al. (2010) argued that a weakening of the zonal jets over the past decades might have contributed to the
average multi-decadal deoxygenation in the ETNA OMZ. In fact, the authors found a multi-decadal oxygen
decrease over the whole depth range between 100 and 1,000 m likely being the result of a weakening of the



wind-driven surface intensified zonal currents (NECC, nNECC) as well as the low baroclinic mode LAZJ.
Multi-decadal changes in the STC or the MOC might have led to long-term changes in the oceanic oxygen
concentration as well (Duteil et al., 2014a), but no relation to oxygen was shown so far for the ETNA.

In the recent decade, oxygen decreased above the OMZ core (shallower than 400 m) and increased below.
Diapycnal supply and meridional eddy supply were assumed to be time invariant in the oxygen budget and time
dependence was included in the residual oxygen supply (cf. Eq. (1)) mainly associated with zonal advection.
First, the slight shoaling of the residual supply in the upper 350 m indicated a weakening and shoaling of the
near-surface circulation resulting in a weakened ventilation of the ETNA. Second, the vertically homogeneous
increase of the residual supply below 400 m suggested a homogeneous strengthening of the circulation over
depth. Obviously, this homogeneous change in the residual supply fits best with the structure of the residual
supply profile itself, whereas temporal variability in the diapycnal or meridional eddy supply would likely have
led to a non-homogeneous change in the oxygen supply. We shall additionally note that almost at the depth of
strongest negative multi-decadal oxygen trend the decadal oxygen trend is zero switching from negative (above)
to positive (below). This depth also coincides with the minimum of the residual oxygen supply separating a
region with strong near surface supply from the region with weaker, depth-independent supply below. The
different vertical structures of the decadal and multi-decadal trends, together with the residual supply profile,
further suggest different ventilation mechanisms at work for the different time scales.

It might be argued that a depth-independent oxygen trend below 400 m could have been induced by a temporal
change of several processes superimposing each other in the oxygen budget. Lateral eddy diffusion has already
been ruled out as the main driver for this decadal trend. Decadal variability in the diapycnal supply, which is
parameterized as a diffusive process given as $-\frac{\partial}{\partial z}\Phi = \frac{\partial}{\partial z}(\rho K_\rho \frac{\partial}{\partial z}O_2)$ (with diapycnal oxygen flux $\Phi$, density $\rho$,
diapycnal diffusivity $K_\rho$ and vertical oxygen gradient $\frac{\partial}{\partial z}O_2$, Fischer et al. (2013)), has not explicitly been
investigated so far. The question is, how much of a change in $K_\rho$ is required to explain the observed trend. The
trend was typically found in the range ±1 μmol kg$^{-1}$ yr$^{-1}$, which is the same order of magnitude as the diapycnal
oxygen supply itself. Hence, $K_\rho$ needed to have varied by a factor 2 to explain this trend. Recent observational
(2008 – 2015) studies quantified $K_\rho$ for the ETNA and found only slight differences between 1.0×10$^{-5}$ m$^2$ s$^{-1}$ to
1.19×10$^{-5}$ m$^2$ s$^{-1}$ (with 95% confidence limits 0.8×10$^{-5}$ m$^2$ s$^{-1}$ and 1.4×10$^{-5}$ m$^2$ s$^{-1}$) in the depth range of 150 to
600 m (Banyte et al., 2012; Fischer et al., 2013; Köllner et al., 2016). Such a marginal variability in strength of
the diapycnal mixing cannot exclusively account for the observed oxygen trend at all – neither for the
deoxygenation in the deep oxycline nor for the oxygenation in the OMZ core. Note also that constant
diffusivities would always act to damp a weakening or strengthening of an oxygen minimum.

So far, the impact of vertical advection on the oxygen concentration in the ETNA was not discussed, as
observationally based estimates of vertical velocity are rare. Even though a robust analysis is impossible, we
roughly estimated the advective vertical oxygen supply by referring to a model study from Pena-Izquierdo et al.
(2015), who came up with a rough vertical structure of the mean vertical velocity in the ETNA. Given their
results, which are of the same order of magnitude as an observationally based estimate from Banyte et al. (2012),
we constructed a vertical velocity profile with upward velocities in the UCW layer (-3×10$^{-7}$ m s$^{-1}$ at 100 m or
26.3 kg m$^{-3}$) and downward velocities in the LCW (1.5×10$^{-7}$ m s$^{-1}$ at 535 m or 27.15 kg m$^{-3}$), and a zero crossing
at the interface of both layers (at 270 m or 26.8 kg m$^{-3}$). We then estimated the advective vertical oxygen supply





by taking the divergence of the advective vertical oxygen flux ($-w\,\partial O_2/\partial z$) and found alternating depth ranges of oxygen gain (roughly $100 - 180$ m and $270 - 430$ m) and oxygen loss (roughly $180 - 270$ m and below 430 m) with magnitudes of up to 0.3 µmol kg$^{-1}$ yr$^{-1}$ (not shown). Even though, this term seems to be up to 5 to 10 times smaller than other oxygen supply terms, its vertical structure at depth of the OMZ is qualitatively similar to the vertical structure of the observed decadal oxygen change (Fig. 9a and Fig 14). This indicates that a decadal change of the advective vertical oxygen supply could additionally contribute to a decadal oxygen change.

**5. Summary and conclusion**

Based on moored, repeat shipboard, Argo float and altimetry observations carried out mainly in the region of the ETNA OMZ, this study explicitly quantified decadal changes of hydrography and oxygen, which were related to decadal changes in the circulation and ventilation. A time dependent oxygen budget of the ETNA was evaluated and decadal to long-term changes of oxygen supply pathways were discussed. The major results are the following:

(1) In the past decade, the ETNA was subject to an accelerated oxygen decrease around the deep oxycline (200 - 400 m) and a large-scale depth-independent oxygen increase below 400 m. This decadal oxygen change pattern was superimposed on the multi-decadal deoxygenation pattern observed in recent studies for the ETNA. While the decadal change in the oxygen distribution was shown to be mainly associated with an upward migration of the OMZ core in density and depth space, the multi-decadal oxygen decrease, that is found above and below the deep oxycline, is associated with a northward shift of the OMZ.

(2) The decadal oxygen change was predominantly negatively correlated with the decadal salinity change in the LCW and IW layers (>300 m) south of the OMZ core. In particular, a large-scale salinity increase was observed for the tropical Atlantic (10°S-10°N) at depth of the deep oxycline potentially going along with a propagation of a positive salinity anomaly from the southern hemisphere across the equator toward the ETNA.

(3) The decadal oxygen decrease and salinity increase in the deep oxycline (200 – 400 m) of the ETNA were associated with local and remote changes in the circulation. A weakening and southward shift of the wind-driven current field between 5°N and 10°N (NECC, nNECC) or a southward shift of the Cape Verde Current between 12°N and 14°N might have directly led to a weakened ventilation of the ETNA. Changes in the Agulhas leakage or the subduction rate in the subtropical South Atlantic might have changed the characteristics of SAW reaching the ETNA.

(4) Below the deep oxycline (>400 m), a likely reason for the intensified ventilation is a strengthening of the LAZJ. They are considered as a major branch of the subsurface mean circulation in the ETNA.

(5) Temporal changes of the different oxygen supply terms, given in the oxygen budget, likely have different impact on the temporal variability of the oxygen distribution. Given the observed decadal oxygen change, zonal advection was associated with a strong decadal change, whereas reasonable changes in lateral and diapycnal diffusion might not have been able setting the amplitude and/or the vertical structure of the oxygen tendency observed for the recent decade.



This study addressed the hydrographic and oxygen variability on intraseasonal to multi-decadal time scales in the
ETNA. Nevertheless, a comprehensive quantitative analysis of the different physical drivers causing the
superimposed variability patterns in oxygen or hydrography, as well as the variability in the oxygen supply
pathways, is still lacking. Changes in oxygen and hydrography might be phase-lagged to changes in the
circulation and longer periods of observational data are necessary in order to capture such relations.

Coupled climate-biogeochemistry models are generally not yet capable of reproducing the mean shape of the
OMZs (Meissner et al., 2005; Cabre et al., 2015) as well as the observed oxygen variability (Stramma et al.,
2012). The lack of a realistically simulated equatorial mean and variable zonal current system is thought to be
one reason for this misrepresentation. Duteil et al. (2014b) showed that an increased model resolution improves
the pattern of the equatorial current system in the tropical Atlantic and correspondingly the oxygen distribution.
Furthermore, they point out that small changes in the strength of the zonal current system have different effects
on oxygen supply and consumption and consequently can affect the oxygen variability. Thus, the simulation of
decadal and multi-decadal oxygen variability might be an even more challenging issue. In an analysis of global
multi-decadal oxygen changes, (Stramma et al., 2012) showed that the observed mean global deoxygenation was
captured by a biogeochemical Earth system model. However, the observed and simulated oxygen change pattern
were negatively correlated suggesting that regional scale changes in oxygen supply and consumption and
responsible processes are so far not reproduced by the model simulations.

Circulation variability can be generally related to climate modes like the Pacific Decadal Oscillation (PDO), the
North Atlantic Oscillation (NAO), the AMOC or the AZM and AMM (Doi et al., 2010; Hormann et al., 2012;
Chang et al., 2006; Marshall et al., 2001; Zhu et al., 2012). Characteristic time scales of these modes project into
circulation variability and consequently may be associated with hydrographic and oxygen variability on similar
time scales (Frölicher et al., 2009; Czeschel et al., 2012; Duteil et al., 2014a). Hence, these modes might be
promising proxies for the description of decadal to multi-decadal oxygen and hydrographic variability.

**Data availability**

The MIMOC climatology is available at http://www.pmel.noaa.gov/mimoc/. Argo float data were collected and
made freely available by the international Argo project and the national programs that contribute to it
(http://doi.org/10.17882/42182). The Argo data was downloaded in March 2016. Aviso altimetry products of
surface geostrophic velocity were downloaded from http://aviso.altimetry.fr/index.php?id=1271 in July 2016
(processing date: 27-Nov-2015). DOI registration for shipboard and mooring data is still in progress and
accessible at https://doi.pangaea.de/10.1594/PANGAEA.869568.

*Competing interests.* The authors declare that they have no conflict of interest.

*Acknowledgements.* This study was funded by the Deutsche Forschungsgemeinschaft as part of the Sonderforschungsbereich
"Climate-Biogeochemistry Interactions in the Tropical Ocean" through several research cruises with RV *L'Atalante*, RV
*Maria S. Merian*, RV *Meteor*, RV *Polarstern*, and by the Deutsche Bundesministerium für Bildung und Forschung (BMBF)
as part of projects NORDATLANTIK (03F0605B, 03F0443B) and RACE (03F0651B). We thank the captains and crew as
well as our technical group for their support with the fieldwork. We thank J. Lübbecke, M. Dengler and T. Fischer for helpful
discussions and R. Kopte for post-processing of the recent ship section data. Further, we thank the international Argo





program and the national programs that contribute to it, which collected the data and made it freely available. The altimeter products were produced by Ssalto/Duacs and distributed by Aviso, with support from Cnes (http://www.aviso.altimetry.fr/duacs/).

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



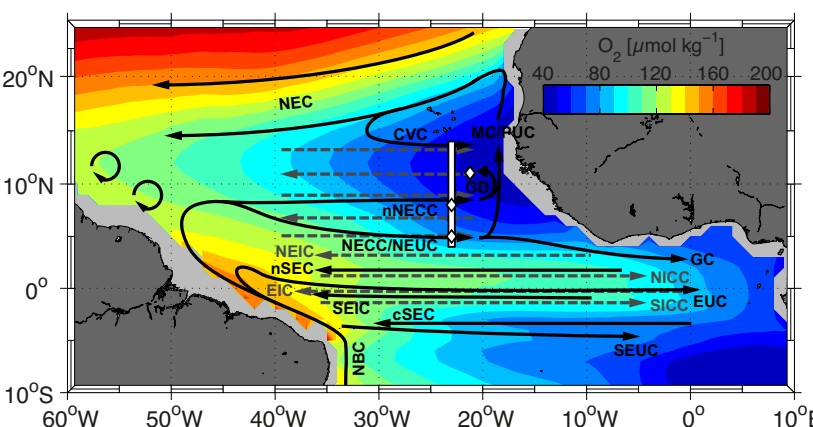

**Figure 1.** Oxygen concentration in µmol kg$^{-1}$ (shaded colors) in the tropical Atlantic at potential density surface 27.1 kg m$^{-3}$ (close to the deep oxygen minimum) obtained from the MIMOC climatology. Superimposed arrows

denote the mean current field (adapted from Brandt et al. (2015); Pena-Izquierdo et al. (2015)). Surface and thermocline (about upper 300 m) currents (black solid arrows) are North Equatorial Current (NEC), Cape Verde Current (CVC), Mauritania Current / Poleward Undercurrent (MC/PUC), Guinea Dome (GD), North Equatorial Countercurrent / North Equatorial Undercurrent (NECC/NEUC), northern branch of the NECC (nNECC), northern and central branch of the South Equatorial Current (nSEC, cSEC), Equatorial Undercurrent (EUC),

South Equatorial Intermediate Current (SEIC), South Equatorial Undercurrent (SEUC) and North Brazil Current (NBC), Current branches at intermediate depth (gray dashed arrows) are latitudinally alternating zonal jets (LAZJ) between 5°N and 13°N, North Equatorial Intermediate Current (NEIC), Equatorial Intermediate Current (EIC) as well as Northern and Southern Intermediate Countercurrent (NICC and SICC). White bar denotes the 23°W section between 4°N and 14°N, which was surveyed with repeat shipboard observations during the recent

decade. White diamonds mark positions of multi-year-long moored observations used in this study.





**Table 1.** Research cruises carried out in the eastern tropical North Atlantic between 4°N and 14°N mainly along the 23°W section between 1999 and 2015. Different columns denote latitude section along mean longitude (lat. / lon.) and maximum profile depth available/used here (depth [m]) for velocity observations performed with vessel-mounted and lowered Acoustic Doppler Current Profilers (vm-ADCP / l-ADCP) as well as of

hydrographic and oxygen observations performed with a conductivity-temperature-depth sonde (CTD/$O_2$). Note that during some research cruises two velocity and/or hydrographic sections were obtained.

| Vessel and cruise (date) | vm-ADCP / l-ADCP | | CTD/$O_2$ | |
|---|---|---|---|---|
| | lat. / lon. | depth [m] | lat. / lon. | depth [m] |
| Thalassa (Jul-Aug 1999) | 4°N-6°N / 23°W | 1000 | - | - |
| Meteor 55 (Oct 2002) | 4°N-10°N / 24°W | 650 | - | - |
| Ron Brown (Jun-Aug 2003) | 4°N-10°N / ~27°W | 1000 | - | - |
| Polarstern Ant XXII/5 (Jun 2005) | 4°N-14°N / 23°W | 300 | - | - |
| Ron Brown (Jun 2006) | 4°N-13.5°N / 23°W | 750 | 4°N-13.5°N / 23.0°W | 1000 |
| | 4°N-14°N / 23°W | 750 | 4°N-14°N / 23.0°W | 1000 |
| Meteor 68/2 (Jun-Jul 2006) | 4°N-14°N / 23°W | 1000 | 4°N-14°N / 23.2°W-22.0°W | 1000 |
| Ron Brown (May 2007) | 4°N-14°N / 23°W | 750 | 4°N-14°N / 23.1°W-22.6°W | 1000 |
| L'Atalante (Feb-Mar 2008) | 4°N-14°N / 23°W | 400 | 4°N-14°N / 23°W | 1000 |
| | 4°N-14°N / 23°W | 1000 | | |
| Maria S. Merian 08/1 (April-May 2008) | 7.5°N-14°N / 23°W | 600 | - | - |
| | 8°N-14°N / 25°W | 800 | | |
| Polarstern Ant XXIV/4 (Apr-May 2008) | 4°N-14°N / 24°W-22°W | 240 | - | - |
| Maria S. Merian 10/1 (Nov-Dec 2008) | 4°N-14°N / 23°W | 650 | 4°N-14°N / 23°W | 1000 |
| Polarstern Ant XXV/5 (May 2009) | 4°N-14°N / 23°W | 250 | - | - |
| Ron Brown (Jul-Aug 2009) | 4°N-14°N / 23°W | 700 | 4°N-14°N / 23°W | 1000 |
| Meteor 80/1 (Oct-Nov 2009) | 4°N-14°N / 23°W | 1000 | 4°N-14°N / 23°W | 1000 |
| | 4°N-14°N / 23°W | 600 | | |




| Polarstern Ant XXVI/1 (Nov 2009) | 4°N-14°N / 23°W | 250 | - | - |
|---|---|---|---|---|
| Meteor 81/1 (Feb-Mar 2010) | 4°N-13°N / 22°W | 1000 | - | - |
| Polarstern Ant XXVI/4 (Apr-May 2010) | 4°N-13.5°N / 23°W | 250 | - | - |
| Ron Brown (May 2010) | 4°N-14°N / 23°W | 1000 | - | - |
| Maria S. Merian 18/2 (May-Jun 2011) | 4°N-5°N / 23°W | 1000 | - | - |
|  | 4°N-14°N / 23°W | 1000 |  |  |
| Maria S. Merian 18/3 (Jun-Jul 2011) | 4°N-14°N / 23°W | 600 | 4°N-14°N / 23°W | 1000 |
| Ron Brown (Jul-Aug 2011) | 4°N-14°N / 23°W | 700 | - | - |
| Maria S. Merian 22 (Oct-Nov 2012) | 4°N-8°N / 23°W | 1000 | 4°N-14°N / 23°W | 1000 |
|  | 4°N-14°N / 23°W | 1000 |  |  |
| Meteor 106 (Apr-May 2014) | 4°N-14°N / 23°W | 1000 | 4°N-14°N / 23°W | 1000 |
| Polarstern PS88.2 (Nov 2014) | 4°N-14°N / 23°W | 1000 | 4°N-14°N / 23°W | 1000 |
| Meteor 119 (Sep-Oct 20 15) | 4°N-14°N / 23°W | 1000 | 4°N-14°N / 23°W | 1000 |




**Table 2.** Moored observations carried out in the eastern tropical North Atlantic between 2009 and 2015. Column 'depth [m]' denotes the instrument depth at the respective mooring. Columns 'O$_2$' and 'T , S' denote the

percentage of available oxygen and hydrographic data, respectively, compared to the total time period. A cross ('x') marks data coverage of better than 99%.

| mooring position (time period) | depth [m] | O$_2$ | T , S |
|---|---|---|---|
| **5°N, 23°W (Nov 2009 - Sep 2015)** | 100 | 87% | x |
| | 200 | 81% | 97% |
| | 300 | 76% | x |
| | 400 | x | x |
| | 500 | 75% | x |
| | 600 | x | x |
| | 700 | 52% | x |
| | 800 | x | 74% |
| **8°N, 23°W (Nov 2009 - Oct 2012)** | 100 | x | x |
| | 200 | x | x |
| | 300 | x | 96% |
| | 400 | x | x |
| | 500 | x | x |
| | 600 | x | 14% |
| | 700 | x | 70% |
| | 800 | x | 98% |
| **11°N, 21.2°W (Nov 2012 - Sep 2015)** | 100 | x | 97% |
| | 200 | x | x |
| | 300 | x | x |
| | 400 | x | x |
| | 500 | x | x |
| | 600 | x | x |
| | 700 | 72% | x |
| | 800 | x | x |





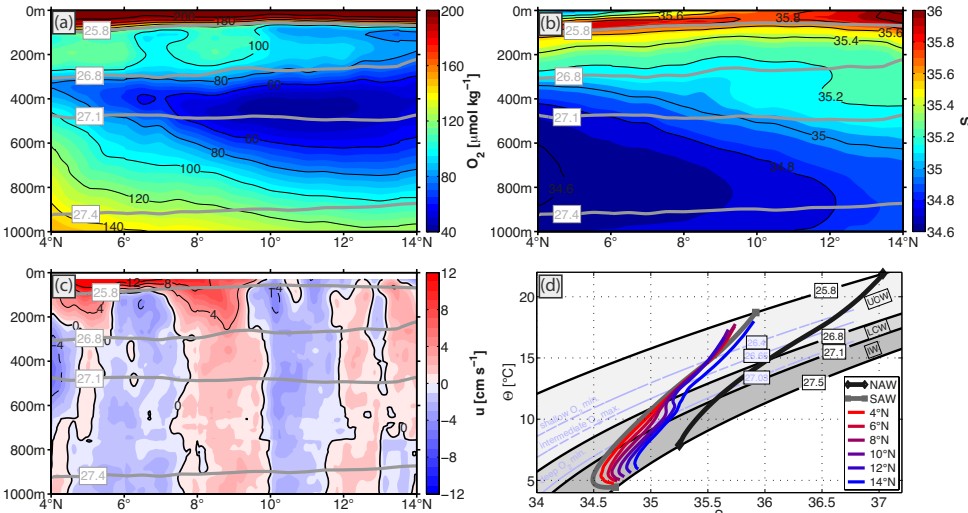

**Figure 2.** Mean sections of (a) oxygen, (b) salinity and (c) zonal velocity along 23°W between 4°N and 14°N.
Black contours denote isolines of the respective mean field. Gray solid contours mark mean surfaces of potential
density in kg m$^{-3}$. Note that no data of mean zonal velocity was derived for the upper 30 m. (d) Mean $\theta$-$S$
characteristics for North Atlantic Water (NAW) and South Atlantic Water (SAW) and for different latitudes
along 23°W (details on reference regimes for NAW and SAW are given in the text). Data from about the upper
100 m of the water column was excluded. Black solid lines denote isopycnal surfaces, which define the different
water mass regimes for UCW (Upper Central Water), LCW (Lower Central Water) and IW (Intermediate
Water). Blue dashed lines denote isopycnal surfaces, which define different oxygen regimes with respect to the
mean vertical oxygen profile in the ETNA.



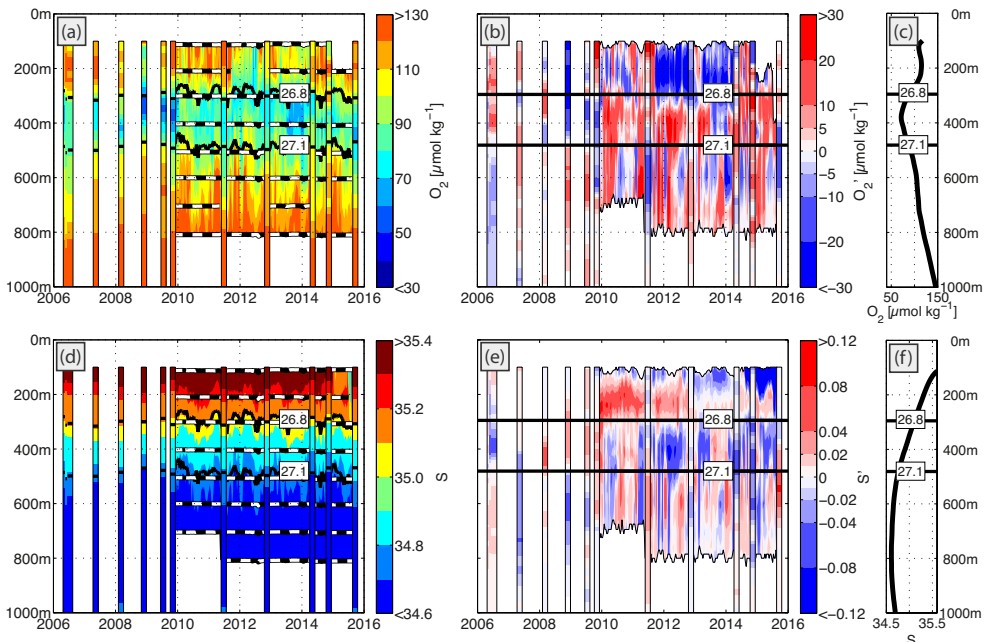

**Figure 3**. Time series of (a) oxygen and (d) salinity in depth space at 5°N, 23°W from moored observations (filled contours) and shipboard observations (vertical bars). Thick black contour lines show isopycnal surfaces (26.8 kg m$^{-3}$ and 27.1 kg m$^{-3}$) and thick black-white dashed lines mark time periods and depths of actual oxygen and salinity measurements from moored observations. (b) and (e) show oxygen and salinity anomalies, respectively, which were calculated on density surfaces with respect to the mean vertical profile from shipboard observations (as given in (c) and (f)) and subsequently projected back onto depth grid. Thick black horizontal lines in (b), (c), (e) and (f) mark the average depth of the two given isopycnal surfaces.





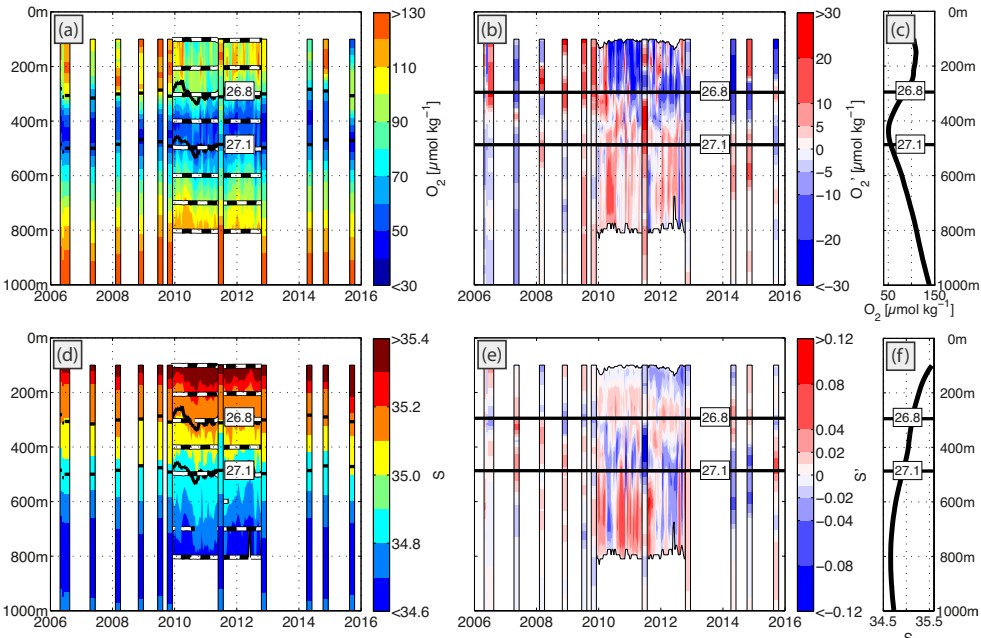

**Figure 4**. Same as Fig. 3, but for 8°N, 23°W.



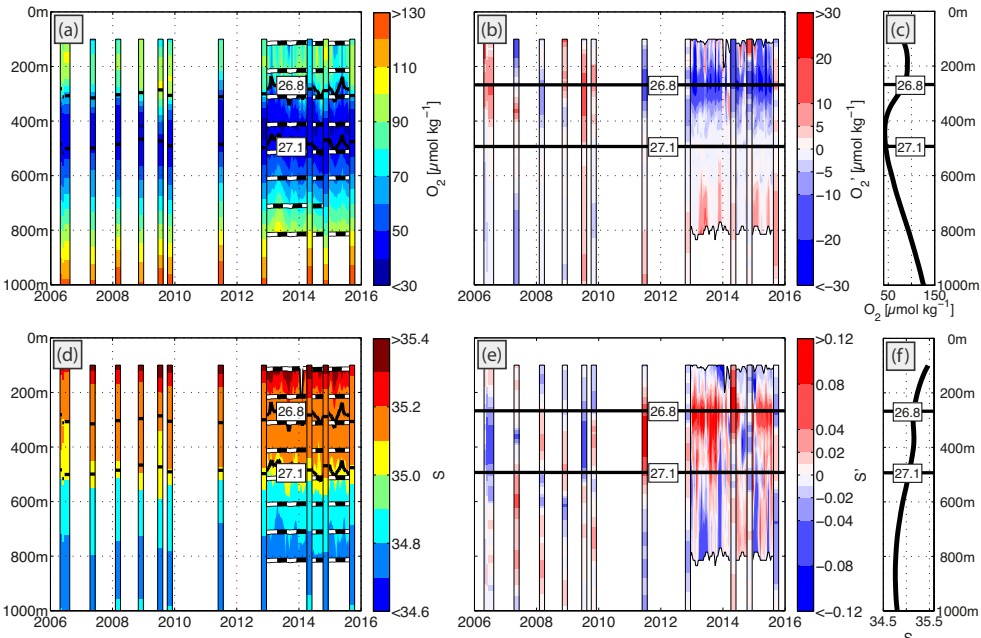

**Figure 5**. Same as Fig. 3, but for 11°N (note the slightly different longitudes for moored (21°W) and shipboard (23°W) observations).





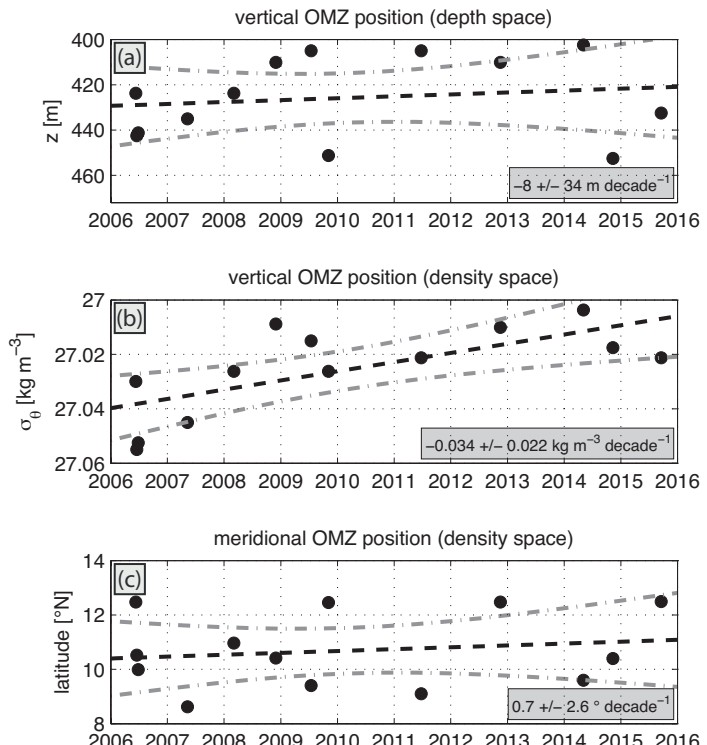

**Figure 6**. Time series (black dots) of the OMZ core position obtained from individual ship sections along 23°W. (a) and (b) show the vertical position in depth and density space. (c) shows the meridional position in density

space (that is similar in depth space). Black dashed line shows the respective linear trend with 95% confidence band (gray dashed-dotted lines).



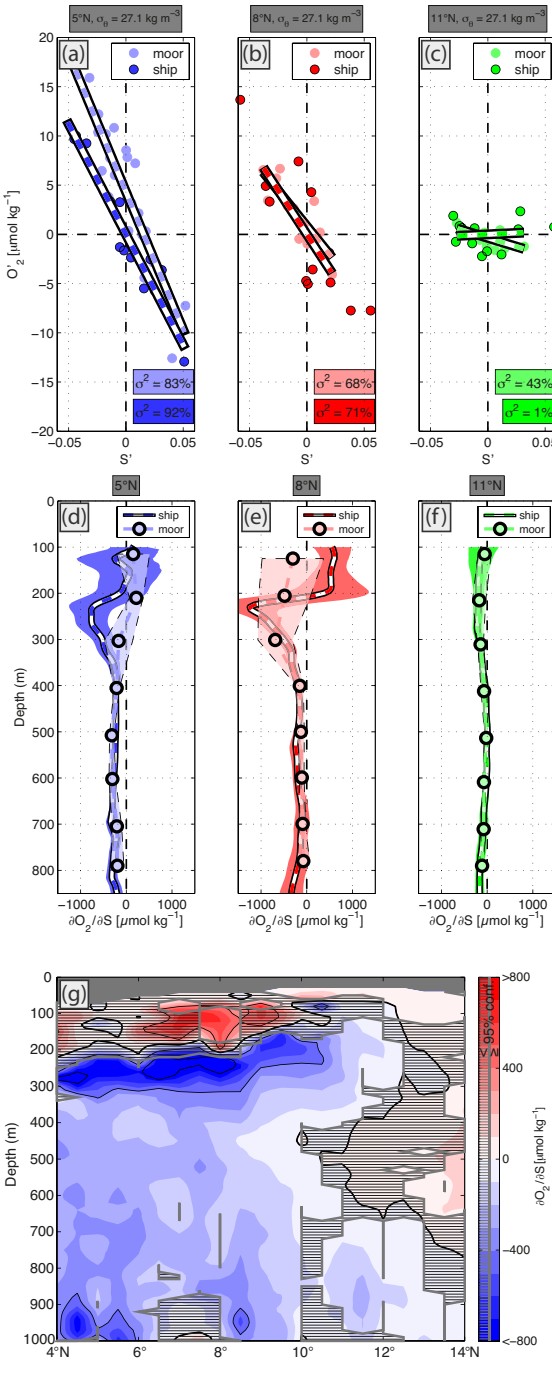

**Figure 7**. Oxygen vs. salinity at the isopycnal surface 27.1 kg m$^{-3}$ from moored (bright colored dots) and shipboard (dark colored dots) observations at (a) 5°N, 23°W, (b) 8°N, 23°W and (c) 11°N, 21°W. Thick colored

dashed lines mark linear fits of oxygen against salinity and percentages in the colored boxes at the bottom mark



the explained variance for the respective observational data set. (d) to (f) show vertical profiles of the slopes of the linear fits of oxygen against salinity with 95% confidence intervals for moored (white-colored dashed line) and shipboard observations (colored dots) with same positions as in (a) to (c). (g) Depth-latitude section (along 23°W) of the slope of the linear fit of oxygen against salinity (filled contours). Gray-hatched areas define non-

significant regimes with respect to 95% confidence.





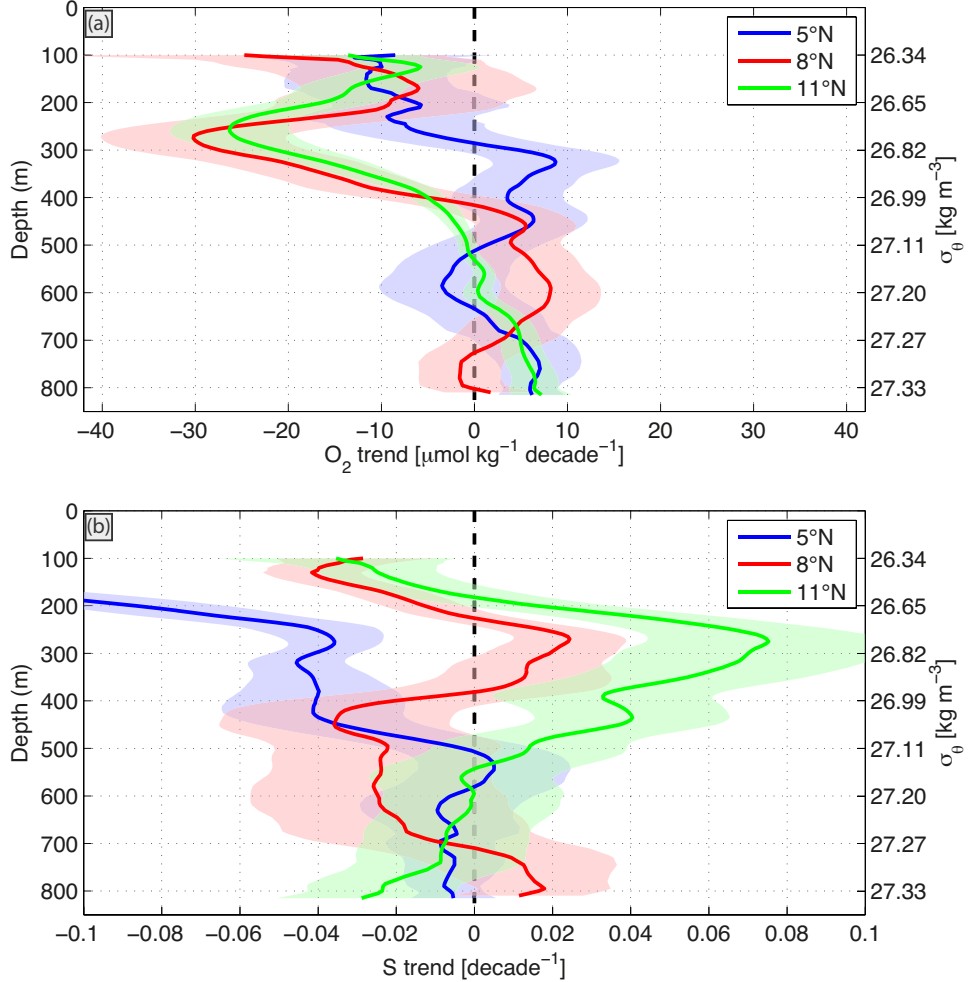

**Figure 8**. Vertical profiles (against depth and density) of the linear trend of (a) oxygen and (b) salinity calculated from the respective time series as a combination of moored and shipboard observations (Fig. 3 to 5) in the ETNA at 5°N, 23°W (blue), 8°N, 23°W (red) and 11°N, 21°W (green). Respective 95% confidence intervals are given as shaded areas. Note that moored and shipboard observations for the latitude position 11°N were at different longitudes 21°W and 23°W, respectively. See text for details of the calculation.





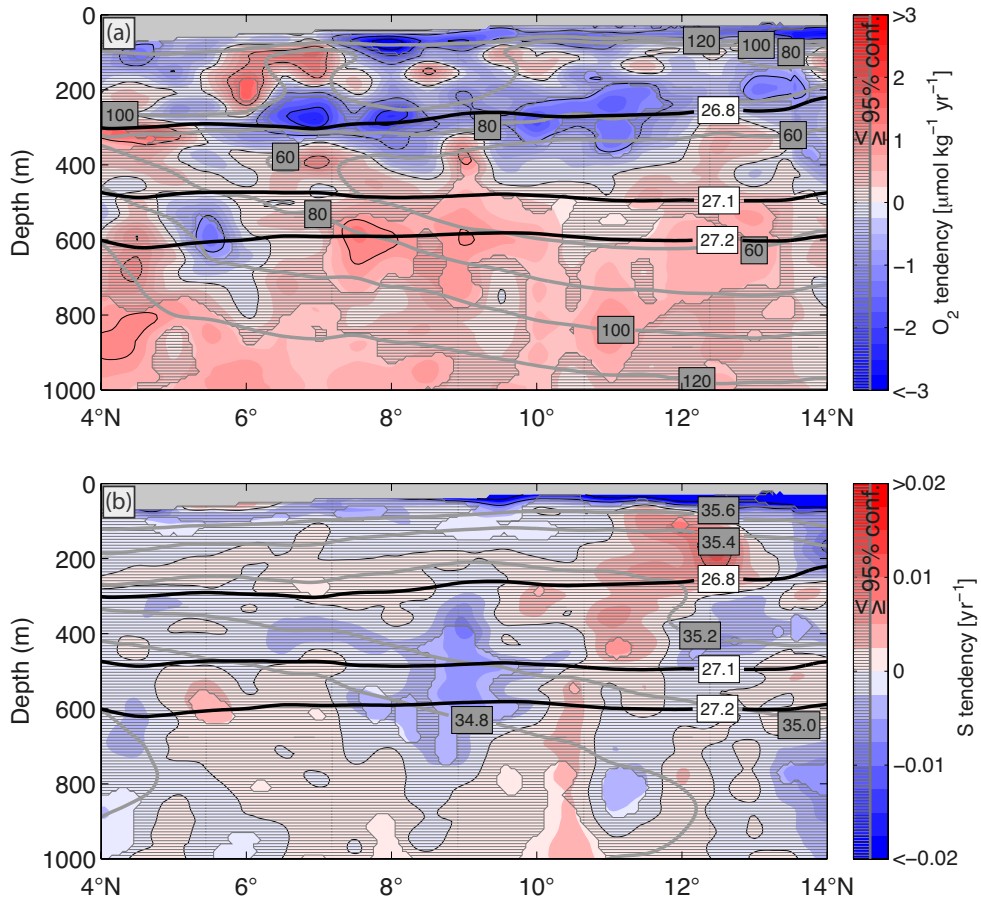

**Figure 9**. Depth-latitude section of the linear trend (filled contours) of (a) oxygen and (b) salinity in the ETNA along 23°W calculated from repeat shipboard observations in the period 2006-2015. The calculation was done on density surfaces with subsequent projection onto depth surfaces. Gray-hatched areas mark non-significant regimes with respect to 95% confidence. Mean fields of oxygen and salinity, respectively, are given as gray contours. Thick black contours define isopycnal surfaces 26.8 kg m$^{-3}$, 27.1 kg m$^{-3}$ and 27.2 kg m$^{-3}$.





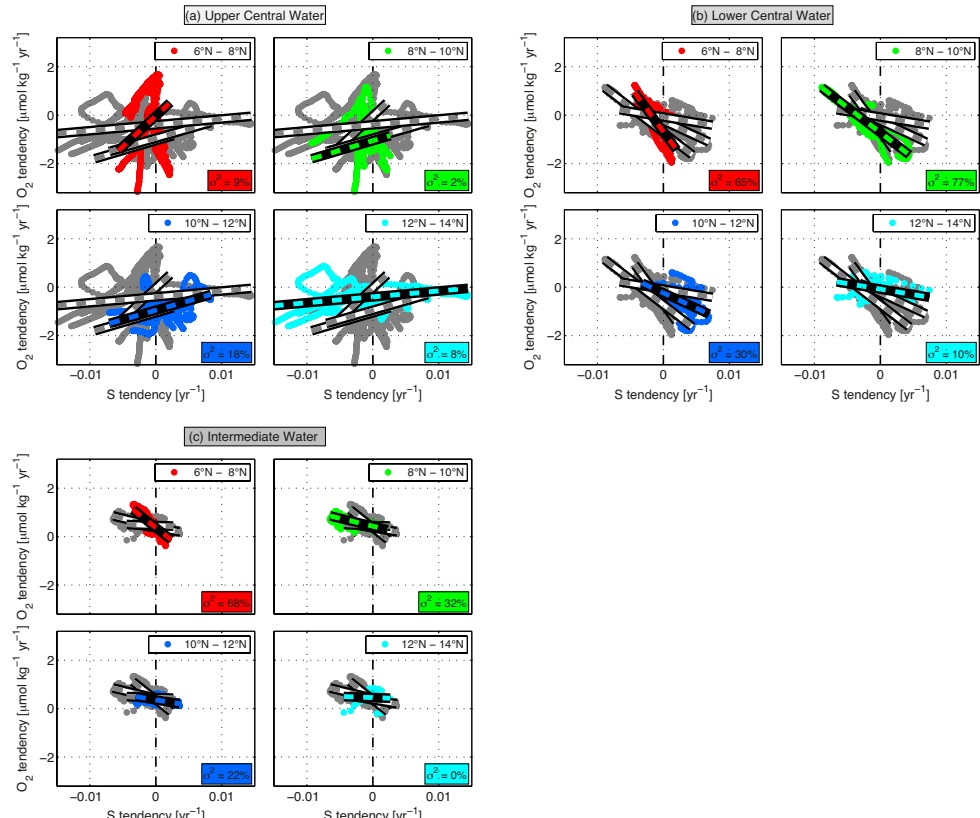

**Figure 10**. Oxygen tendency vs. salinity tendency (dots) at 6°N-8°N (red), 8°N-10°N (green), 10°N-12°–N (dark
blue) and 12°N-14°N (light blue) for (a) Upper Central Water, (b) Lower Central Water and (c) Intermediate
Water. Gray dots in each subpanel denote oxygen vs. salinity tendency for the respective other latitude regimes.
Colored-dashed lines mark the linear fit oxygen tendency against salinity tendency for the respective latitude
regimes. Gray-dashed lines denote the linear fits of the respective other latitude regimes. Percentages in the
colored boxes define the explained variance.





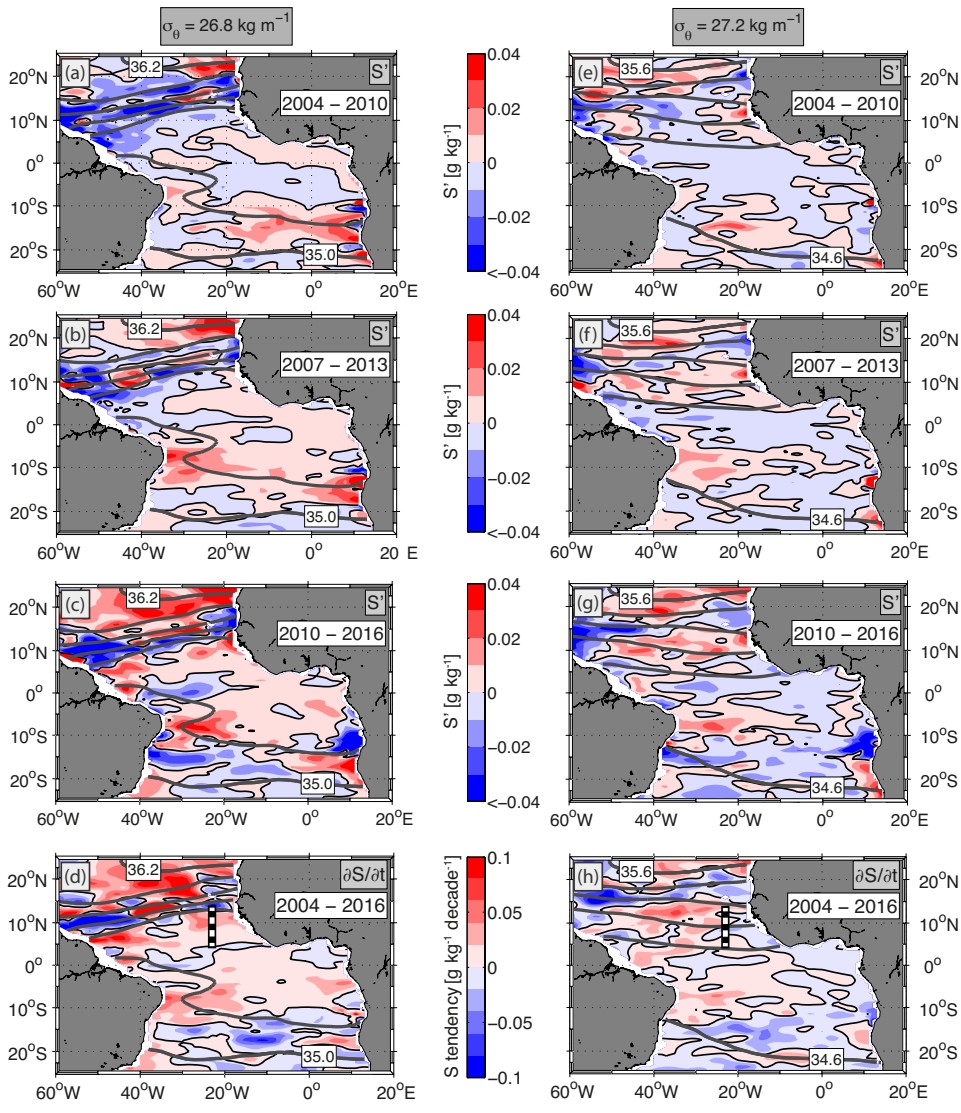

**Figure 11**. Salinity anomalies (filled contours) in the tropical Atlantic at the isopycnal surface 26.8 kg m$^{-3}$ from Argo float observations for the period (a) 2004-2010, (b) 2007-2013 and (c) 2010-2016. (d) Salinity trend (filled contours) calculated from all salinity anomalies between 2004 and 2016. Black-white dashed line marks the 23°W section between 4°N and 14°N for reference. Gray contours in panels (a) to (d) define the mean salinity distribution. (e) – (h): Same as (a) – (d), but for isopycnal surface 27.2 kg m$^{-3}$.





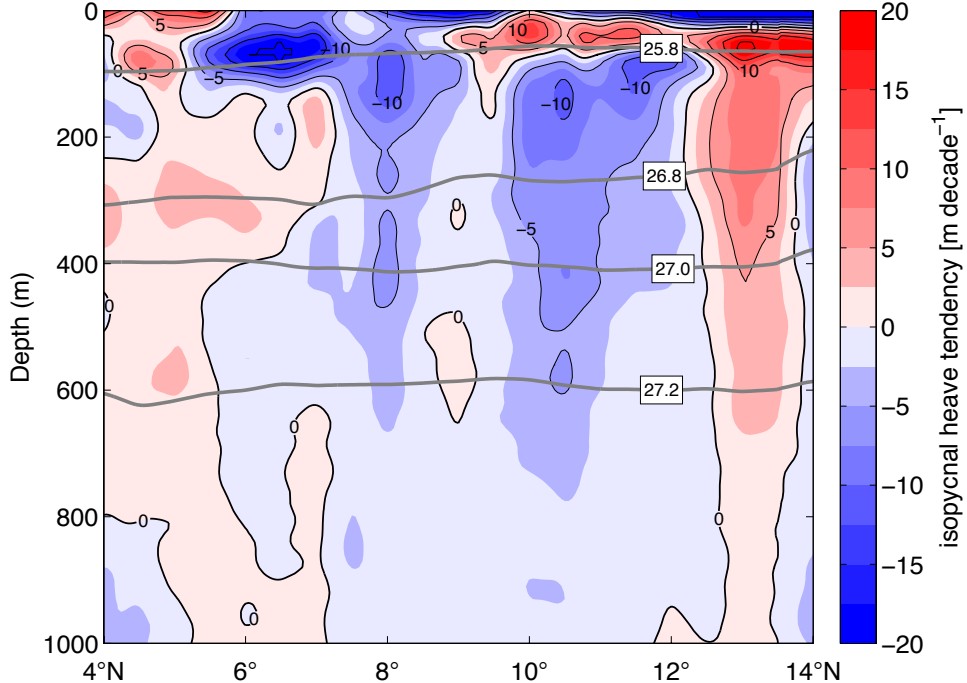

**Figure 12**. Depth-latitude section of the isopycnal heave tendency (filled contours) from shipboard observations along 23°W for the period 2006-2015 (positive values denote upward migration of isopycnal surfaces - see text for details of the calculation). Gray contours define average isopycnal surfaces.





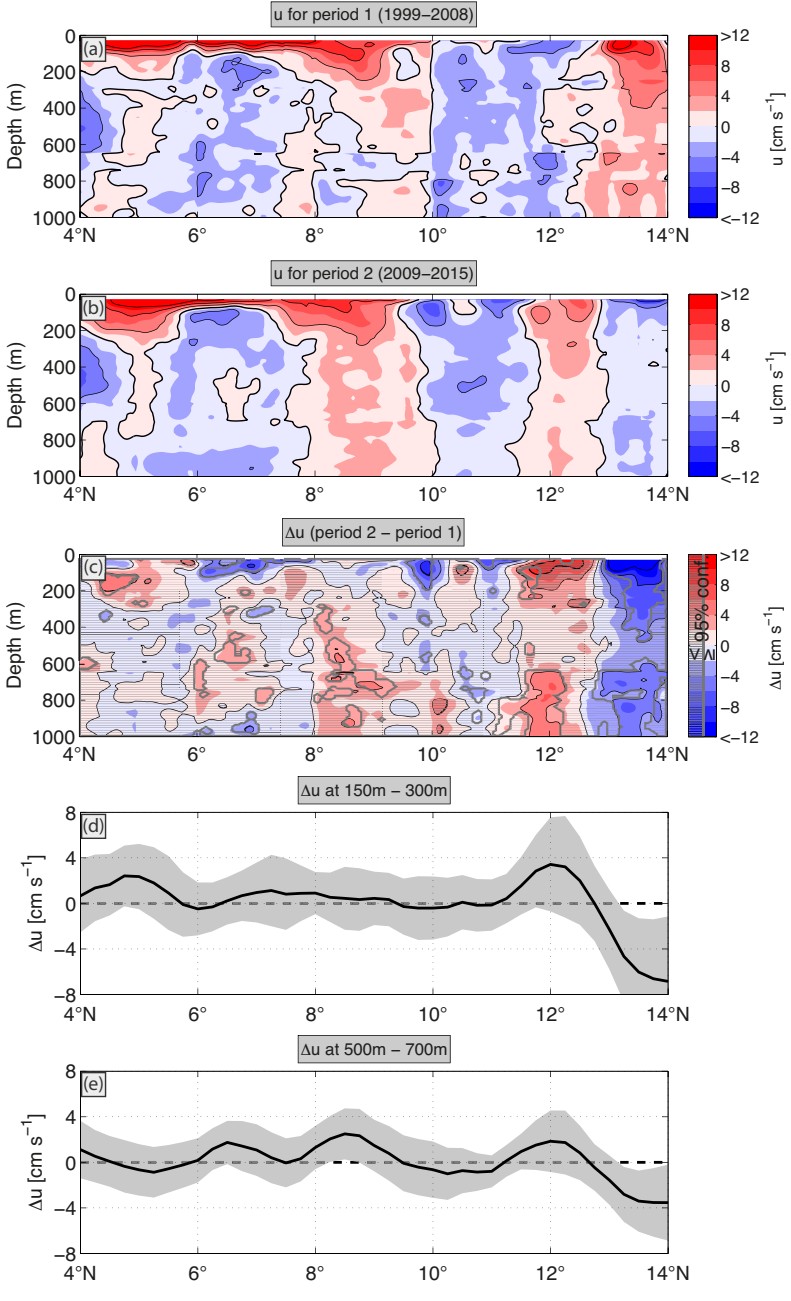

**Figure 13**. Depth-latitude section of mean zonal velocity from shipboard observations along 23°W for (a) period 1999 - 2008 and (b) period 2009 - 2015. (c) Depth-latitude section of the difference of zonal velocity between the aforementioned two periods '(b) minus (a)'. Gray hatching defines non-significant regimes with respect to 95% confidence. (d) Vertical average of (c) for the depth range 150 - 300 m (black line). Gray-shaded area marks 95% confidence band. (e) Same as (d) but for the depth range 500 - 700 m.

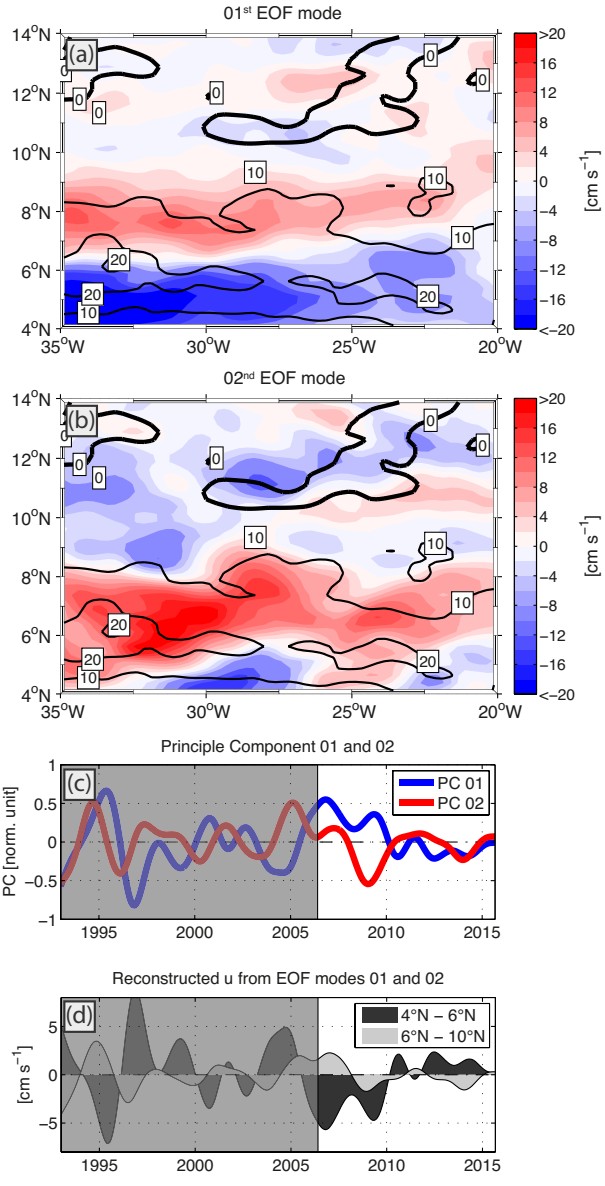

**Figure 14**. (a) First and (b) second EOF mode of the filtered (mean and seasonal cycle removed and subsequently 2-yr low-pass filtered – see text for details) surface zonal geostrophic velocity from AVISO (filled contours; positive eastward) for the tropical North Atlantic. Black contours define isolines of mean surface zonal geostrophic velocity from AVISO in cm s$^{-1}$ (positive eastward). (c) Principle components 01 and 02 corresponding to the first two EOF modes given in (a) and (b). (d) Time series of eastward velocity (averaged between 35°W and 20°W for the latitude bands 4°N-6°N and 6°N-10°N) reconstructed from the first two EOF modes, given in (a) and (b), and corresponding principle components in (c). White boxes (right parts of panels



(c) and (d)) mark the time period (2006 – 2015) of shipboard observations used for the estimate of the decadal
oxygen and salinity trend section in Fig. 9 (cf. Table 1). Gray boxes (left parts) mark the time period before.



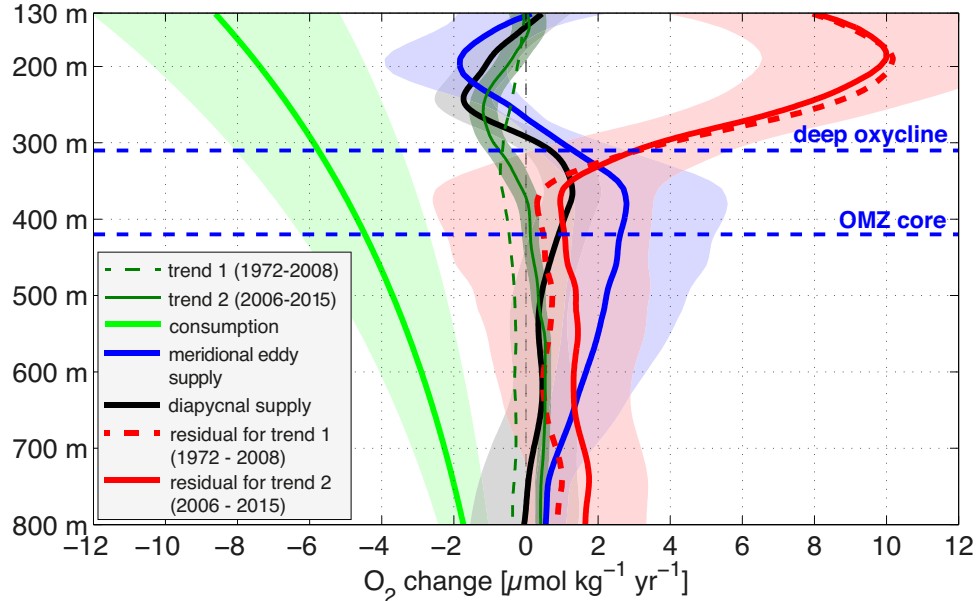

**Figure 15**. Oxygen budget for the ETNA as an average between 6°N and 14°N (see Eq. (1) and corresponding text in Sect. 2.5, 3.4 and 4.3 for details). The terms consumption, meridional eddy supply and diapycnal supply were kept constant in time, whereas two different oxygen trends for the periods 1972-2008 (multi-decadal trend: dashed green line) and 2006-2015 (decadal trend: solid green line) were implemented in the budget. The corresponding residual supply for both periods (multi-decadal: dashed red line – decadal: solid red line) was calculated by assuming a balance between all respective terms.