# Peer review of "Decadal oxygen change in the eastern tropical North Atlantic"

_Ocean Science, 2016_

## Referee Comment (RC1) · Anonymous Referee #1 · 13 Feb 2017

Review of the manuscript os-2016-102 Decadal oxygen change in the eastern tropical North Atlantic Johanes Hahn et al.

The manuscript present an study on decadal variability of Eastern North Tropical Atlantic (ETNA) Oxygen Minimum Zone (OMZ) from in situ measurement (including mooring and repeated shipboard section). The authors show decadal negative trend of dissolved oxygen (DO) within the upper OMZ (200-400 m), while increase of DO is shown within lower OMZ layer (400-1000 m). This is attributed to a southward shift of the wind-driven zonal circulation in the North Tropical Atlantic, which supplies the ETNA in oxygen enriched water including South Atlantic subtropical Waters. Isopycnal salinity/spiciness is also used as proxy to infer DO concentration variability (negative correlation in the ETNA) in the broader context of the upper South and Tropical Atlantic circulation.

The manuscript presents an interesting multi-sensor approach to document the variability of last decade of the ETNA OMZ. This is a valuable contribution to better understand the complex variability and dynamics of the DO in this region, which as been intensively investigate since ten years by this group. However, the manuscript is a little bit too long and lack of concision. Some Figures are poor and not easy to read (that make them not convincing). Also some supplementary analyses could be valuable to strengthen the results (oxygen and salinity transports).

Suggestion of the reviewer: moderate revision.

Comments/Questions:

I.108: typo : 'Theta-S'

I.217: The advection which is the key mechanism is estimated as residual. Is it possible (why if not) to make an estimate of such a term from data at hand? (see comment below)

Section 3.2: This section could be more effective if restructured : The observations of interannual variability and decadal trend need to be first introduce (Fig3-4-5-8-9), then mechanism of passive advection and correlation with spiciness should be discussed (Fig. 6-7-10-11).

I.296-298: The DO increase of the lower layer may be barely significant. Figures 3 and 4 does not show such a tendency.

Fig.3-5: is it possible to group this figures to allow an better overview of the ship and mooring data set at hand. Also, please improve this figure to make them more readable (plot the time series in row panels for example to elongate them and improve the readability). Why it is it the mean isopycnal that is represented on the right panels and not the full isopycnal depth time series? Also, It seems that the seasonal to interannual variability may be important : discontinuity between mooring measurements and shipboard sections are questionable. Please better comment this point. OSD
I. 308-309: Maybe introduce that the ventilation process by passive tracer circulation appear to be a significant mechanism in the lower layer of OMZ. However, I would first introduce the Fig. 8 and 9 before trying to correlate between S and DO. Indeed, it seems that the link between spiciness and DO less robust, than the observed trends of both S and DO (Fig. 7 and 10), and need to be discussed in the larger context of the STC along with the Fig. 11.

Fig. 8-9: Try to group these figures

Section 3.3:

I.372-380: This is a striking results of the paper. It would be more convincing if DO and S transport were calculated. Also divergence of these transports between  $23^{\circ}W$  and African coast (if a good assumption) could help to estimate the variability of zonal advection of DO and S?

I.381-402: What is the driver of such meridionnal shift of the surface currents ? (wind curl?)

OSD

---

## Referee Comment (RC2) · Anonymous Referee #2 · 3 Mar 2017

The manuscript updates the observed decadal trend of Dissolved Oxygen (DO) of Eastern Tropical North Atlantic (ETNA) using both shipboard and moored measurements, as well as satellite Sea Surface Height (SSH) and Argo float profiles. The authors found that the most recent decadal trend is characterized by a di-pole structure with decreasing DO in the 100-400m depth range and increasing trend below between 400m and 800m. This is in contrast to the previously identified multidecadal trend declining more uniformly from 100m to 800m. Results suggest that low frequency DO changes are more complicated than previously thought. Authors attribute the DO trend in the most recent decade to a southward shift of wind driven circulation in the upper layer and strengthening of Latitudinally Alternating Zonal Jets (LAZJ) in the deeper part. The study presents an interesting analysis of in situ measurements in the ETNA Oxygen Minimum Zone (OMZ) and should be publishable after addressing the following concerns. A minor revision is recommended.

Major Concern: The presentation is not easy to follow mainly because evidence shown in several of the figures are not robust and noisy. Some of the following specific comments may help to improve the manuscript.

Specifics:

1) Line 108: "q-S" typo here?

2) Line 230, residual term in eq.1, "[5] . . . composite of mean advection, zonal eddy diffusion as well as submesoscale processes": Mean advection shall be separately estimated by using the shipboard and mooring data.

3) Much of the significant changes, DO or salinity (S), are identified on density surfaces 26.8 and 27.2, but presented in difference-maps of two time periods (e.g., Fig.9). These two density surfaces happen to be the upper and lower bound of OMZ (Fig. 2). Even though isopycnal heaving has been ruled out as the main reason for observed variations, time series of moored DO and S, together with data points from shipboard measurements, shall be presented in the data section or 3.2 on these two density surfaces and 27.0 (corresponding to the center of OMZ).

4) Line 300, "OMZ core . . . was estimated . . .by . . . center of 1%..": How many data points, corresponding to the 1%, are used for estimating the position of OMZ core? Will using 5% of the data change the conclusion?

5) Fig.7: How the moored timeseries data are represented by data points, every 90 days? The colored dots in the slope line can be confused with the data points. Consider using thin straight lines to show the linear fit?

6) Fig.9b and Fig8b: Most of the S changes in Fig.9b are insignificant. There is also the large inconsistency between Fig.8b and Fig.9b at 5N. Why is so? Add another sub-figure based on Argo data? Fig.9a and b are noisy, not easy to identify the significant changes. Will smoothing help?

7) Fig.10: What is the purpose to show the insignificant correlations in Fig.10. There are too many subpanels. Only show significant ones?

8) Fig.11a-c and e-f: too noisy to show the propagation signals. Would Hovmoller diagrams, either along 23W or zonal average, do better job?

9) Fig.13 vs. Fig.14: The most striking changes of zonal currents are between 12N and 14N in the shipboard ADCP data (Fig.14), while the most significant changes in geostrophic currents are between 4N and 8N in Fig.13. Why is this difference?
* * *

---

## Author Comment (AC1) · 23 May 2017

**Authors' response on manuscript "Decadal oxygen change in the eastern tropical North Atlantic"**

Johannes Hahn[1], Peter Brandt[1], Sunke Schmidtko[1], Gerd Krahmann[1]

[1]GEOMAR Helmholtz Centre for Ocean Research Kiel, Düsternbrooker Weg 20, 24105 Kiel, Germany

*Correspondence to*: Johannes Hahn (jhahn@geomar.de)

Dear Editor,

we would like to thank you and the two anonymous reviewers for the positive, but thorough evaluation of the manuscript. We appreciate the fruitful comments, which help to improve the manuscript. We have structured our response to the referees' comments as follows: The individual comments from the referees are given in black and are directly followed by the authors' response (AR) given in blue color. The AR not only contains the answer to the reviewer's comment, but also contains a brief description about what was changed in the manuscript ($\rightarrow$ '*manuscript changes*'). All figures given in this authors' response letter are referenced with AR-Fig. ##. All figures from the manuscript that are referred to in the authors' response letter are referenced with Fig. ## (or with 'former Fig. ##' when referring to figures in the published discussion paper).

**Anonymous Referee #1**

Review of the manuscript os-2016-102 Decadal oxygen change in the eastern tropical North Atlantic Johanes Hahn et al.

The manuscript present an study on decadal variability of Eastern North Tropical Atlantic (ETNA) Oxygen Minimum Zone (OMZ) from in situ measurement (including mooring and repeated shipboard section). The authors show decadal negative trend of dissolved oxygen (DO) within the upper OMZ (200-400 m), while increase of DO is shown within lower OMZ layer (400-1000 m). This is attributed to a southward shift of the wind-driven zonal circulation in the North Tropical Atlantic, which supplies the ETNA in oxygen enriched water including South Atlantic subtropical Waters. Isopycnal salinity/spiciness is also used as proxy to infer DO concentration variability (negative correlation in the ETNA) in the broader context of the upper South and Tropical Atlantic circulation.

The manuscript presents an interesting multi-sensor approach to document the variability of last decade of the ETNA OMZ. This is a valuable contribution to better understand the complex variability and dynamics of the DO in this region, which as been intensively investigate since ten years by this group. However, the manuscript is a little bit too long and lack of concision. Some Figures are poor and not easy to read (that make them not

convincing). Also some supplementary analyses could be valuable to strengthen the results (oxygen and salinity transports).

Suggestion of the reviewer: moderate revision.

AR: Thank you very much for the general statement and the positive evaluation of the manuscript. According to the reviewer's comments, we generally did the following changes in the manuscript:

i) We calculated a rough estimate of the lateral advective oxygen supply as well as its temporal change. Even though this is only a qualitative estimate, the vertical profile generally agreed with our results in the oxygen budget. We used this result to strengthen our argumentation that changes in the lateral advective oxygen supply may have substantially contributed to the observed decadal oxygen changes.

ii) We removed detailed descriptions of data analysis methods from Sect. 3 and included them in Sect. 2 in order to improve the readability and comprehensibility of Sect. 3.

iii) We reordered the figures in the result section Sect. 3.2 in order to improve the structure and the flow in the manuscript (Fig. 3 – Fig. 9). The text in Sect. 3.2 was restructured and edited according to the figure order. Some figures, that were poor in content or not easy to read, were removed or edited/regrouped, in particular a better view on the comparison of shipboard and moored observations and float observations.

iv) We included a schematic (Fig. 15) in order to illustrate oxygen and ventilation changes and the associated upward migration of the ETNA OMZ.

In the following specific answers to all reviewer's comments are given including according changes in the manuscript.

Comments/Questions:

l.108: typo : 'Theta-S'

AR: Done. Changed to ' $\theta$-$S$ '.

l.217: The advection which is the key mechanism is estimated as residual. Is it possible (why if not) to make an estimate of such a term from data at hand? (see comment below)

AR: It is quite a challenge to derive a consistent and quantitative estimate of the advective supply solely based on observations. Such an estimate would be only reasonable, if all surfaces (sections) of a well-defined ,ocean box' were measured quasi simultaneously (e.g. as a combination of repeat ship sections and moored observations complemented by other measurement platforms such as floats or gliders) in order to determine all influxes/outfluxes

to/from this box. This cannot be achieved with the existing data set available for this study. However, we followed the reviewer's suggestion and qualitatively estimated the advective supply and its tendency directly from observations in order to strengthen our discussion about the physical drivers of the decadal oxygen change.

For this rather qualitative estimate, we defined a box between 23°W and 15°W and between 6°N and 14°N. We assumed a balanced mass flux through the 23°W section, whereas fluxes through all other boundaries are negligible. This is a very strong assumption, but nevertheless we believe that this represents the effect of the mean current field in this regime to first order of magnitude.

The lateral advective oxygen supply was estimated by calculating the integral of eastward and westward fluxes through 23°W and subsequently dividing the result by the volume of the assumed box as given above. Indeed, the vertical structure of the mean lateral advective oxygen supply as well as of its decadal change (AR-Fig. 1) is qualitatively in agreement with the oxygen budget terms (vertical structure of the residual supply and decadal oxygen change, respectively).

[Figure]

*AR-Fig. 1: Vertical profiles of the mean zonal advective oxygen supply across 23°W (black) and the lateral advective oxygen supply tendency (red) calculated from shipboard observations along 23°W and referred to the box regime 23°W-15°W and 6°N-14°N. Gray shading defines double the standard deviation of the advective oxygen supply and red shading is the 95% confidence band of the advective oxygen supply tendency. (This figure is now Fig. 14 in the manuscript)*

*Manuscript changes:* We have included a passage with a detailed description about the qualitative estimate of the advective oxygen supply and its respective trend in the manuscript's discussion section (ll. 1206–1239). We also included figure AR-Fig. 1 (Fig. 14 in the manuscript) and we embedded a qualitative discussion of the results.

Section 3.2: This section could be more effective if restructured : The observations of interannual variability and decadal trend need to be first introduce (Fig3-4-5-8-9), then mechanism of passive advection and correlation with spiciness should be discussed (Fig. 6-7-10-11).

AR: We have reordered the figures following the reviewer's suggestion. Moreover, the presentation of the time series was simplified to improve comprehensibility (Fig. 3 – 5).

*Manuscript changes:* The order of the figures was restructured as given in the following. The manuscript (mainly Sect. 3.2) was changed accordingly to the figure order.

1) We changed the figures of the mooring time series (former Fig. 3, 4, 5). Now: Fig. 5, 4 and 3. Instead, we plotted time series as averages over two depth regimes (200m-400m and 500m-800m) in order to improve the presentation and comparison of shipboard and moored observations. We also included the trend estimate from moored and shipboard observations together and removed the figure with vertical profiles of the decadal trend (former Fig. 8). The latter change was done according to a specific comment by reviewer 2.

2) We kept former Fig. 9 (trend estimates from shipboard observations), but we included an additional subpanel with an estimate of the salinity trend based on float observations. Now: Fig. 6.

3) We removed former Fig. 10 (correlation of oxygen and salinity trend) and included a table with oxygen and salinity trends for different boxes averaged over latitude and depth. Now: Table 3. This shall improve the structure and the thread of the manuscript.

4) We kept former Fig. 7, as this helps to better understand the correlation of oxygen and salinity as well as the relation of the respective trends (given in Fig. 6 and Table 3). Now: Fig. 7.

5) We kept former Fig. 11 showing the salinity trends from float observations at density surfaces 26.8 and 27.2. Now: Fig. 8.

6) We kept former Fig. 6 about the OMZ location. Now: Fig. 9.

l.296-298: The DO increase of the lower layer may be barely significant. Figures 3 and 4 does not show such a tendency.

AR: The combination of moored and shipboard observations shall generally give an overview about the existing data set. Indeed, there is a lot of variability existing on different time scales which makes it challenging in ad-hoc interpreting the time series. However, particularly for 11°N (as well as for 8°N) the combined presentation of both data sets strengthens our argumentation, since moored observations show negative oxygen anomalies in the upper OMZ layer (above 400 m) and positive oxygen anomalies in the lower OMZ layer (below

400 m). The now plotted depth average of the time series over the upper and the lower depth layer shall provide a more detailed view on the time scales existing in the time series (Fig. 3 to 5).

130  Most striking is the meridional section of the decadal oxygen change based on shipboard observations, where a significant average oxygen decrease (5.9 +/- 3.5 µmol kg$^{-1}$ decade$^{-1}$) was found in the upper layer between 200 and 400 m and a significant average oxygen increase (4.0 +/- 1.6 µmol kg$^{-1}$ decade$^{-1}$) was found in the lower layer between 400 and 1000 m, both as an average between 6°N and 14°N. The spatially coherent structures in both depth layers further strengthen the robustness of this estimate.

135  We agree with the reviewer's argumentation, that shipboard and moored observations at 5°N, 23°W don't show a decadal oxygen change in the respective two depth layers. Here, intraseasonal, seasonal and interannual variability plays an important role, likely driven by tropical instability waves as well as variability in the zonal currents (NECC/NEUC).

*Manuscript changes:* We have rewritten the passage in the manuscript about oxygen time
140  series and the decadal oxygen tendency from shipboard observations (ll. 473-481). Now, depth averaged time series are shown in the manuscript (Fig. 3 to Fig. 5), also according to a suggestion from reviewer 2. In ll. 482-612, significant oxygen tendencies with confidence intervals are given (i) as an average between 6°N and 14°N and (ii) as box averages (table 3). The decadal salinity tendency was additionally calculated from float obervations along 23°W
145  (Fig. 6c) in order to make this estimate more robust. The text passage was adapted accordingly in ll. 613-618.

Fig.3-5: is it possible to group this figures to allow an better overview of the ship and mooring data set at hand. Also, please improve this figure to make them more read- able (plot the time series in row panels for example to elongate them and improve the readability). Why it is it
150  the mean isopycnal that is represented on the right panels and not the full isopycnal depth time series? Also, It seems that the seasonal to interannual variability may be important : discontinuity between mooring measurements and shipboard sections are questionable. Please better comment this point.

AR: Fig. 3, 4 and 5 were completely renewed. Instead showing the full depth range, depth
155  averages of oxygen and salinity anomaly time series were plotted for two depth layers (200-400m and 500-800m). The anomaly time series were evaluated on isopycnal surfaces in order to remove the effect of isopycnal heave. For a more intuitive representation, we converted the results back onto depth grid and calculated the depth average. Thus, a particular depth can be associated as the mean depth for a specific isopycnal surface. The mean depth of the two
160  depth layers corresponds to the isopycnals 26.8 and 27.2.

Intraseasonal to interannual variability becomes more important at the southern rim of the OMZ (we stated this in more detail in ll. 473-481 of the manuscript). Particularly at 5°N, no significant decadal changes in oxygen or salinity could be generally observed and

intraseasonal to interannual variability is dominant, likely driven by tropical instability waves as well as variability in the zonal currents (NECC/NEUC). This can be directly seen in the mooring time series of oxygen anomaly and salinity anomaly, where dominant time scales of positive and negative anomalies are between a month and several years. The strong inconsistency between estimates of decadal change in salinity from shipboard observations on the one hand (former Fig. 9b) and a combination of moored and shipboard observations on the other hand (former Fig. 8b) was mainly in the upper 200m. We revisited again the salinity time series at 100m for the last mooring period of the 5°N mooring. Indeed, the corresponding sensor likely had a bad conductivity cell after about 1/5 of the mooring period. This data was removed. Note that at 5°N, salinity fluctuations on a time scale of about the whole decade could be observed at depths between 100 and 300 m.

At 8°N and 11°N, intraseasonal to interannual variability is less pronounced compared to 5°N as can be directly seen from moored observations. Decadal changes become more important. Thus, estimates of decadal changes become more robust. However, moored observations at 8°N and 11°N are too short to use them alone for an estimate of decadal changes in oxygen and salinity. Shipboard observations provide a reasonable alternative to estimate decadal changes due to the reduced variability on intraseasonal to interannual time scales.

*Manuscript changes:* Fig. 3, 4 and 5 were completely changed and all time series anomalies were plotted as depth averages for two depth layers (200-400m and 500-800m). However, the time series anomalies were calculated in density space. More details about the time series were given in ll. 473-481.

l. 308-309: Maybe introduce that the ventilation process by passive tracer circulation appear to be a significant mechanism in the lower layer of OMZ. However, I would first introduce the Fig. 8 and 9 before trying to correlate between S and DO. Indeed, it seems that the link between spiciness and DO less robust, than the observed trends of both S and DO (Fig. 7 and 10), and need to be discussed in the larger context of the STC along with the Fig. 11.

AR: If the reviewer meant that dissolved oxygen and salinity are not robustly correlated in the upper 200-300m of the water column as well as around the OMZ core, we agree. However, at depth of the lower Central Water and Intermediate Water, both variables correlate well south of the OMZ core with an average slope of about $\frac{\partial O_2}{\partial S}$ = -255 +/- 140 µmol/kg (we additionally stated this in the manuscript in l. 623). A qualitatively similar relation is also shown for the oxygen vs. salinity trend south of the OMZ core. Close to the OMZ core, oxygen vs. salinity correlation is not given at all. However, as mentioned already in the manuscript, a direct relation among salinity changes and water mass or circulation changes cannot be drawn.

*Manuscript changes:* We included one sentence in the beginning of the respective paragraph to make clear, that oxygen and salinity changes are analyzed in order to study the effect of physical ventilation processes (ll. 619-620).

We followed the reviewer's suggestion and reordered former Fig. 6 to 10. Former Fig. 9 (now: Fig. 6) is introduced before former Fig. 6, 7 and 10. Former Fig. 8 and 10 were removed. A detailed overview about the figure order is given in the response to the reviewer's 3$^{rd}$ comment.

205     We included a quantitative estimate of the slope $\frac{\partial O_2}{\partial S}$ = -255 +/- 140 μmol/kg in the manuscript in l. 623 in order to underline the significant correlation of oxygen and salinity south of the OMZ core and below the deep oxycline.

There were already described several remote processes, which might have lead to salinity changes in the eastern tropical North Atlantic (see ll. 990–996 in the manuscript). In addition,
210     we included a paragraph in Sect. 4.2 in order to state the potential impact of these processes on oxygen changes in the ETNA (ll. 1003-1074).

Fig. 8-9: Try to group these figures

AR/*Manuscript changes*: Former Fig. 8 was removed and instead, the trends were included in the depth averages of oxygen and salinity anomaly time series (Fig. 3, 4, 5) - according also to
215     reviewer 2). Former Fig. 9 (Now: Fig. 6) was kept, but an additional subpanel was included showing the estimate of the decadal salinity change from float observations.

l.372-380: This is a striking results of the paper. It would be more convincing if DO and S transport were calculated. Also divergence of these transports between 23°W and African coast (if a good assumption) could help to estimate the variability of zonal advection of DO
220     and S ?

AR: We followed the reviewer's suggestion and qualitatively estimated the oxygen flux divergence (oxygen supply) due to zonal advection. As this point was already mentioned earlier by the reviewer, details are given above in the response to the reviewer's 2$^{nd}$ comment.

The estimate of the zonal advection was solely based on oxygen. This was not done for
225     salinity, since no salinity budget was calculated for the ETNA.

l.381-402: What is the driver of such meridionnal shift of the surface currents ? (wind curl?)

AR: Yes, we strongly suspect that changes in the large-scale wind stress curl may have driven (Sverdrup) circulation changes in the upper few hundred meters in the eastern tropical North Atlantic. The quantitative investigation of this mechanism goes beyond this study.

230     *Manuscript changes:* The suspected relation between (Sverdrup) circulation changes an the large-scale wind stress curl is pointed out at two defined positions in the manuscript. (i) ll. 1082-1083 (Sect. 4.2: discussion): here we specified our description with ,... wind-driven gyres of the tropical North Atlantic ...'. (ii) ll. 1293-1294 (Sect. 5: Summary and conclusion):

here we specifically included the speculation ,... - which are likely related to changes in the large-scale wind-stress curl - …'

**Anonymous Referee #2**

The manuscript updates the observed decadal trend of Dissolved Oxygen (DO) of Eastern Tropical North Atlantic (ETNA) using both shipboard and moored measurements, as well as satellite Sea Surface Height (SSH) and Argo float profiles. The authors found that the most recent decadal trend is characterized by a dipole structure with decreasing DO in the 100-400m depth range and increasing trend below between 400m and 800m. This is in contrast to the previously identified multidecadal trend declining more uniformly from 100m to 800m. Results suggest that low frequency DO changes are more complicated than previously thought. Authors attribute the DO trend in the most recent decade to a southward shift of wind driven circulation in the upper layer and strengthening of Latitudinally Alternating Zonal Jets (LAZJ) in the deeper part. The study presents an interesting analysis of in situ measurements in the ETNA Oxygen Minimum Zone (OMZ) and should be publishable after addressing the following concerns. A minor revision is recommended.

Major Concern: The presentation is not easy to follow mainly because evidence shown in several of the figures are not robust and noisy. Some of the following specific comments may help to improve the manuscript.

AR: Thank you very much for the general statement and the positive evaluation of the manuscript. According to the reviewer's comments, we generally did the following changes in the manuscript:

i) We improved the structure and the flow in the manuscript, particularly in the result section Sect. 3. Detailed descriptions of data analysis methods were removed from Sect. 3 and included in Sect. 2 in order to improve the readability and comprehensibility of Sect. 3. Some figures, that were poor in content or not easy to read, were removed (former Fig. 8 and 10), other figures were edited or regrouped (former Fig. 3 to 11). The text (particularly in Sect. 3.2) was restructured and edited according to the figure order.

ii) In particular, we completely renewed the figures about the mooring time series (former Fig. 3 to 5). These Figures (Now: Fig. 5, 4 and 3) show depth averages of the anomaly time series for two depth layers (200-400m and 500-800m) to improve the view on the consistency between moored and shipboard observations.

iii) We calculated a rough estimate of the lateral advective oxygen supply as well as its temporal change. Even though this is only a qualitative estimate, the vertical profile generally agreed with our results in the oxygen budget. We used this result to strengthen our argumentation that changes in the lateral advective oxygen supply may have substantially contributed to the observed decadal oxygen changes.

270    iv) We included a schematic (Fig. 15) in order to illustrate oxygen and ventilation changes and the associated upward migration of the ETNA OMZ.

In the following specific answers to all reviewer's comments are given including according changes in the manuscript.

Specifics:

275    1) Line 108: "q-S" typo here?

AR: Changed to ' $\theta$-$S$ '.

2) Line 230, residual term in eq.1, "[5] . . . composite of mean advection, zonal eddy diffusion as well as submesoscale processes": Mean advection shall be separately estimated by using the shipboard and mooring data.

280    AR: It is quite a challenge to derive a consistent and quantitative estimate of the advective supply solely based on observations. Such an estimate would be only reasonable, if all surfaces (sections) of a well-defined ‚ocean box' were measured quasi simultaneously (e.g. as a combination of repeat ship sections and moored observations complemented by other measurement platforms such as floats or gliders) in order to determine all influxes/outfluxes

285    to/from this box. This cannot be achieved with the existing data set available for this study. However, we followed the reviewer's suggestion and qualitatively estimated the advective supply and its tendency directly from observations in order to strengthen our discussion about the physical drivers of the decadal oxygen change.

For this rather qualitative estimate, we defined a box between 23°W and 15°W and between

290    6°N and 14°N. We assumed a balanced mass flux through the 23°W section, whereas fluxes through all other boundaries are negligible. This is a very strong assumption, but nevertheless we believe that this represents the effect of the mean current field in this regime to first order of magnitude.

The lateral advective oxygen supply was estimated by calculating the integral of eastward and

295    westward fluxes through 23°W and subsequently dividing the result by the volume of the assumed box as given above. Indeed, the vertical structure of the mean lateral advective oxygen supply as well as of its decadal change (AR-Fig. 1_2) is qualitatively in agreement with the oxygen budget terms (vertical structure of the residual supply and decadal oxygen change, respectively).

[Figure]

*AR-Fig. 1_2: Vertical profiles of the mean lateral advective oxygen supply across 23°W (black) and the lateral advective oxygen supply tendency (red) calculated from shipboard observations along 23°W and referred to the box regime 23°W-15°W and 6°N-14°N. Gray shading defines double the standard deviation of the advective oxygen supply and red shading is the 95% confidence band of the advective oxygen supply tendency. (This figure is now Fig. 14 in the manuscript)*

*Manuscript changes:* We have included a passage with a detailed description about the qualitative estimate of the advective oxygen supply and its respective trend in the manuscript's discussion section (ll. 1206–1239). We also included figure AR-Fig. 1_2 (Fig. 14 in the manuscript) and we embedded a qualitative discussion of the results.

3) Much of the significant changes, DO or salinity (S), are identified on density surfaces 26.8 and 27.2, but presented in difference-maps of two time periods (e.g., Fig.9). These two density surfaces happen to be the upper and lower bound of OMZ (Fig. 2). Even though isopycnal heaving has been ruled out as the main reason for observed variations, time series of moored DO and S, together with data points from shipboard measurements, shall be presented in the data section or 3.2 on these two density surfaces and 27.0 (corresponding to the center of OMZ).

AR/*manuscript changes*: We have calculated depth averages of the time series anomalies for two depth layers in order to explicitly show amplitude and time scales of oxygen and salinity fluctuations both from moored and shipboard observations. The two depth layers (i) 200-400m and (ii) 500-800m show the observed oxygen decrease in the upper layer and the oxygen increase in the lower layer. The depth averaged time series anomalies for the three mooring positions are shown in Fig. 3, 4 and 5 of the manuscript.

4) Line 300, "OMZ core . . . was estimated . . .by . . . center of 1%..": How many data points, corresponding to the 1%, are used for estimating the position of OMZ core? Will using 5% of
325 the data change the conclusion?

AR: The 1% area of lowest oxygen was calculated based on all ship sections, which were extracted in the latitude range 8°N-14°N and in the density (depth) range of about 27.0 – 27.06 kg m$^{-3}$ (400m-460m). The 1% area corresponds to 8 data points. For a robustness check, we have redone the analysis by defining the 5% area (about 42 points per ship section)
330 and we ended up with similar results, as shown in AR-Fig. 2.

*Manuscript changes:* We adapted the text (ll. 736–746) and the figure (now: Fig. 9) in the manuscript and now refer our analysis to the 5% area of lowest oxygen. We additionally stated that choosing the 1% area gave similar results. We didn't mention explicitly the number of data points, since this is depending on the grid of the processed ship section data.

[Figure]

335

*Fig. 2: Vertical and meridional migration of OMZ core. Left panels: former Fig. 6 from manuscript (i.e. 1% area of lowest oxygen between 8°N and 14°N). Right panels: new figure in manuscript (Fig. 9, i.e. 5% area of lowest oxygen was used).*

5) Fig.7: How the moored timeseries data are represented by data points, every 90 days? The
340 colored dots in the slope line can be confused with the data points. Consider using thin straight lines to show the linear fit?

AR/*manuscript changes:* Yes, the data points shown in the scatter plot represent the 90 day median values of the mooring time series. However, according to both reviewers' general statements, we decided to remove panels a to c, since they don't substantially contribute to the
345 main question/message ('How do dissolved oxygen and salinity correlate?') of corresponding paragraphs. The new figure is shown in Fig. 7 in the manuscript.

6) Fig.9b and Fig8b: Most of the S changes in Fig.9b are insignificant. There is also the large inconsistency between Fig.8b and Fig.9b at 5N. Why is so? Add another sub- figure based on Argo data? Fig.9a and b are noisy, not easy to identify the significant changes. Will smoothing help?

AR: We agree that most of the salinity changes estimated from shipboard observations (former Fig. 9b) are locally insignificant with respect to a 95% confidence interval. The results are very variable. We also see the inconsistency at 5°N between estimates from shipboard observations on the one hand (former Fig. 9b) and a combination of moored and shipboard observations on the other hand (former Fig. 8b). Exemplarily, in about the upper 300m, shipboard observations suggest substantial salinity fluctuations over the whole decade (compare new Fig. 5 in the manuscript) with a minimum in the beginning and the end of the period as well as a maximum in between. The period of moored observations only captures the branch of the salinity decrease. Moreover, we revisited again the salinity time series at 100m for the last mooring period for the 5°N mooring. Indeed, the corresponding sensor likely had a bad conductivity cell after about 1/5 of the mooring period. This data was removed.

In contrast, we have to disagree with the reviewer's statement that also the section of oxygen change is too noisy for interpretation. Even though parts of the estimated changes are insignificant, we found valuable regimes with significant (95% level) oxygen change. These regimes become even more trustworthy due to the large and homogeneous pattern of oxygen decrease and increase (this is already stated in the manuscript in ll. 482-612).

A direct comparison of the oxygen change pattern with the estimated salinity change pattern from shipboard observations might be taken with care. In order to come up with a more robust estimate of the decadal salinity change and thus with a more robust comparison of the oxygen changes, we directly estimated the decadal salinity change from float observations (similar period as for ship section data, i.e. 2006 – 2015). AR-Fig. 3 shows the decadal oxygen and salinity change from shipboard observations (as before) as well as the decadal salinity change from float observations along 23°W.

[Figure]

AR-Fig. 3: (a) Decadal oxygen tendency and (b) decadal salinity trend tendency from shipboard observations between 4°N and 14°N (similar to former Fig. 9 in the manuscript). (c) Decadal salinity tendency from float observations along 23°W between 4°N and 14°N. All observations are from the period 2006-2015 (this figure is now Fig. 6 in the manuscript).

380    The decadal salinity change erstimated from float observations is more significant than the corresponding estimate from the very limited number of shipboard observations. However, both estimates show a positive salinity tendency between about 6°N and 13°N at about 200 - 300 m (isopycnal 26.8). Below 400 m, both estimates show a good coherence of alternating regimes of salinity increase and decrease along the section. A significant salinity
385    decrease is found between about 7°N and 10°N.

The salinity increase above 350 m generally goes along with a coherent oxygen decrease north of 6°N. Below 400 m, the salinity decrease between about 7°N and 11°N goes along with a significant oxygen increase. We think, this consideration better shows the anticorrelated relation of oxygen and salinity at both depth layers. We may conclude that
390    these changes are caused by only one major process. In contrast, the incoherent structures of salinity increase and decrease below 400 m might have been caused by several processes (remote or local changes).

*Manuscript changes:*

1) In ll. 473-481, we introduced the different time scales of oxygen and salinity variability in
395    more detail.

2) Details on the robustness of the decadal oxygen tendency along 23°W are given in ll. 482-612 of the manuscript.

3) The decadal salinity change was additionally estimated from float observations and is shown as an additional subpanel (Fig. 6c). The text was adapted accordingly (ll. 613-618).

400    7) Fig.10: What is the purpose to show the insignificant correlations in Fig.10. There are too many subpanels. Only show significant ones?

AR/*manuscript changes*: We agree with the reviewer. We decided to remove the whole figure. Instead we included a table (Table 3), which shows the average decadal oxygen and salinity tendency for different latitude boxes (two degrees width) at the Central Water (200-
405    300m) and Intermediate Water (500-800m) layer, respectively. Oxygen tendency was estimated from shipboard observations. Salinity tendency was estimated from float observations. We think, that this representation supports the understanding about significant regimes of decadal changes in oxygen and salinity and the relation of both. The text was adapated accordingly (ll. 630-633).

410    8) Fig.11a-c and e-f: too noisy to show the propagation signals. Would Hovmoller diagrams, either along 23W or zonal average, do better job?

AR: Literature shows (e.g. Rabe et al. 2008; Biastoch et al. 2009; Kirchner et al. 2009; Lübbecke et al. 2015 – references are also given in the manuscript) that water mass pathways from the subtropical South Atlantic to the tropical and subtropical North Atlantic are due to

415 the Subtropical Cell and the Meridional Overturning Circulation via the energetic western boundary current system. Hence, Hovmoller diagrams along 23°W are not adequate in representing such propagation signals. Hovmoller diagrams of zonal averages don't show a continuous propagation of salinity anomalies from the southern to the northern hemisphere, either (AR-Fig. 4). Indeed, a reasonable definition of water mass pathways (e.g. definition of

420 a streamfunction) would be necessary in order to follow the propagation of these signals. We speculate that Argo float observations were too sparse (particularly in the narrow and energetic western boundary current system) in order to resolve a continuous pathway of northward propagating salinity anomalies.

*Manuscript changes:* We have rewritten the according paragraph (ll.982-989) and included
425 the above described expectation and speculation with regard to the Argo float observations.

[Figure]

*AR-Fig. 4: Hovmoller diagrams of zonally averaged (40°W-10°W) salinity anomalies from Argo float observations. Left panel: isopycnal surface 26.8. Right panel: isopycnal surface 27.2.*

  9) Fig.13 vs. Fig.14: The most striking changes of zonal currents are between 12N and 14N in
430 the shipboard ADCP data (Fig.14), while the most significant changes in geostrophic currents are between 4N and 8N in Fig.13. Why is this difference?

AR: We suspect different reasons for this difference.

First, the zonal (and also meridional) extent as well as the amplitude of the coherent variability pattern of the surface currents is larger between 4°N and 10°N than between 10°N
435 and 14°N. Thus, the EOF analysis of the altimetry observations dominantly reflects the velocity variability between 4°N and 10°N. In contrast, shipboard observations are only given along 23°W and cannot capture the zonal extent of the current field.

Second, the eastward current between 12°N and 14°N (Cape Verde Current) has its core below the surface as shown in the mean velocity section along 23°W (Fig. 2c in the
440 manuscript; see also Brandt et al. 2015 or Pena-Izquierdo et al. 2015 – references are given in the manuscript). The maximum variability of this current is likely at the subsurface as well.

Third, shipboard observations represent only snapshots of the strongly variable current field. Hence, there is likely an aliasing effect in the variability analysis due to the limited number of shipboard observations.

445    However, a detailed variability analysis of the Cape Verde Current goes beyond the scope of this study, but we see this as a worthwhile approach for future studies.

*Manuscript changes:* We included the three above mentioned speculations in the manuscript (ll. 1091-1099) to state this difference between shipboard and altimetry observations.

**Decadal oxygen change in the eastern tropical North Atlantic**

Johannes Hahn[1], Peter Brandt[1], Sunke Schmidtko[1], Gerd Krahmann[1]

[revised manuscript text omitted]